# Congenital heart disease detection by pediatric electrocardiogram based deep learning integrated with human concepts

Jintai Chen [1,17], Shuai Huang[2,3,4,17], Ying Zhang[3,5,17], Qing Chang[6,7,17], Yixiao Zhang[6,8], Dantong Li[2,3,4], Jia Qiu[3,5], Lianting Hu[2,3,4], Xiaoting Peng[2,3,4], Yunmei Du[9,10], Yunfei Gao[11,12], Danny Z. Chen [13], Abdelouahab Bellou[14,15] ✉, Jian Wu[1,16] ✉ & Huiying Liang [2,3,4] ✉

Early detection is critical to achieving improved treatment outcomes for child patients with congenital heart diseases (CHDs). Therefore, developing effective CHD detection techniques using low-cost and non-invasive pediatric electrocardiogram are highly desirable. We propose a deep learning approach for CHD detection, CHDdECG, which automatically extracts features from pediatric electrocardiogram and wavelet transformation characteristics, and integrates them with key human-concept features. Developed on 65,869 cases, CHDdECG achieved ROC-AUC of 0.915 and specificity of 0.881 on a real-world test set covering 12,000 cases. Additionally, on two external test sets with 7137 and 8121 cases, the overall ROC-AUC were 0.917 and 0.907 while specificities were 0.937 and 0.907. Notably, CHDdECG surpassed cardiologists in CHD detection performance comparison, and feature importance scores suggested greater influence of automatically extracted electrocardiogram features on CHD detection compared with human-concept features, implying that CHDdECG may grasp some knowledge beyond human cognition. Our study directly impacts CHD detection with pediatric electrocardiogram and demonstrates the potential of pediatric electrocardiogram for broader benefits.

Congenital heart disease (CHD) is one of the most common type of birth defects and a major cause of children's morbidity and mortality[1,2]. Early and accurate identification of affected pediatric patients is crucial for timely intervention and effective surgical outcomes[3–6]. However, commonly used examination methods, such as transthoracic echocardiography (TTE), X-ray, cardiac magnetic resonance imaging (MRI), and dual-source CT examinations, are operationally complex, time-consuming and costly, and heavily dependent on the evaluation of experienced cardiologists[7]. Unfortunately, the delayed diagnosis is prevalent[8] (even for the critical cases[9]), which results in sub-optimal clinical intervention[10–14], especially in low- and middle-income regions[15–20]. A study in a low-income country has demonstrated that the delay diagnosis rate can be up to 85.1%[21].

In general, CHDs are caused by structural abnormalities such as holes and leaky valves, which change the electrocardiovectors and can present abnormal manifestations in electrocardiogram (ECG) signals theoretically[22–24]. In this context, the surface ECG can offer insights into cardioelectric activity that could be helpful for the detection of CHD patients due to its affordable price and high effectiveness. It has been partly observed that CHD is associated with some particular manifestations on adult ECG signals[25–32], suggesting that evaluating the abnormal ECG waveforms could provide clues for the detection of

underlying heart defects. However, previous research mostly focused on the correlations between CHD and adult ECG, which did not offer an immediate benefit for pediatric CHD interventions[33]. So far, there have been only a few studies on CHD diagnosis using pediatric ECG signals, which brings challenges as well as opportunities for innovative discovery in the field of ECG analysis.

Recent advances in deep learning (DL) have demonstrated cardiologist-level and reliable performances on ECG analysis[34–36], including identifying features not typically recognized by human experts. Earlier investigations have demonstrated that DL models can derive benefits not only from automatically extracted features of ECG waveform data[35–37] but also from conceptual features utilized by human experts[38–42] and features obtained through wavelet transformation[43,44]. However, few approaches integrated all these different feature types in an end-to-end DL architecture to conduct automatic, efficient, and optimal fusion. Moreover, scant attention has been given to the development of DL models for the detection of CHD, particularly on a large-scale pediatric ECG dataset[45–49].

In this work, we present an end-to-end deep neural network-based approach for pediatric ECG cases, called Congenital Heart Disease diagnosis via Electrocardiogram (CHDdECG). CHDdECG optimally integrates multiple input feature types including raw ECG-waveform data, human-concept features, and wavelet features, in order to make direct probabilistic predictions for CHD. The workflow of our study is illustrated in Fig. 1. First, potential pediatric patients underwent several examinations, primarily consisting of transthoracic echocardiography and electrocardiogram, in accordance with the European Society of Cardiology Guidelines for CHD[50]. In certain cases, additional tests may have been used at the discretion of the attending doctors. The doctors carefully analyzed all the examination results and subsequently determined the final diagnostic outcomes. Next, our CHDdECG used only pediatric ECG data to identify CHD cases, by integrating features automatically extracted from ECG-waveform data and wavelet features with human concept features. It was developed using 65,869 pediatric ECG cases of young children in the age of 2.12 ± 1.50 (year), and evaluated on an internal

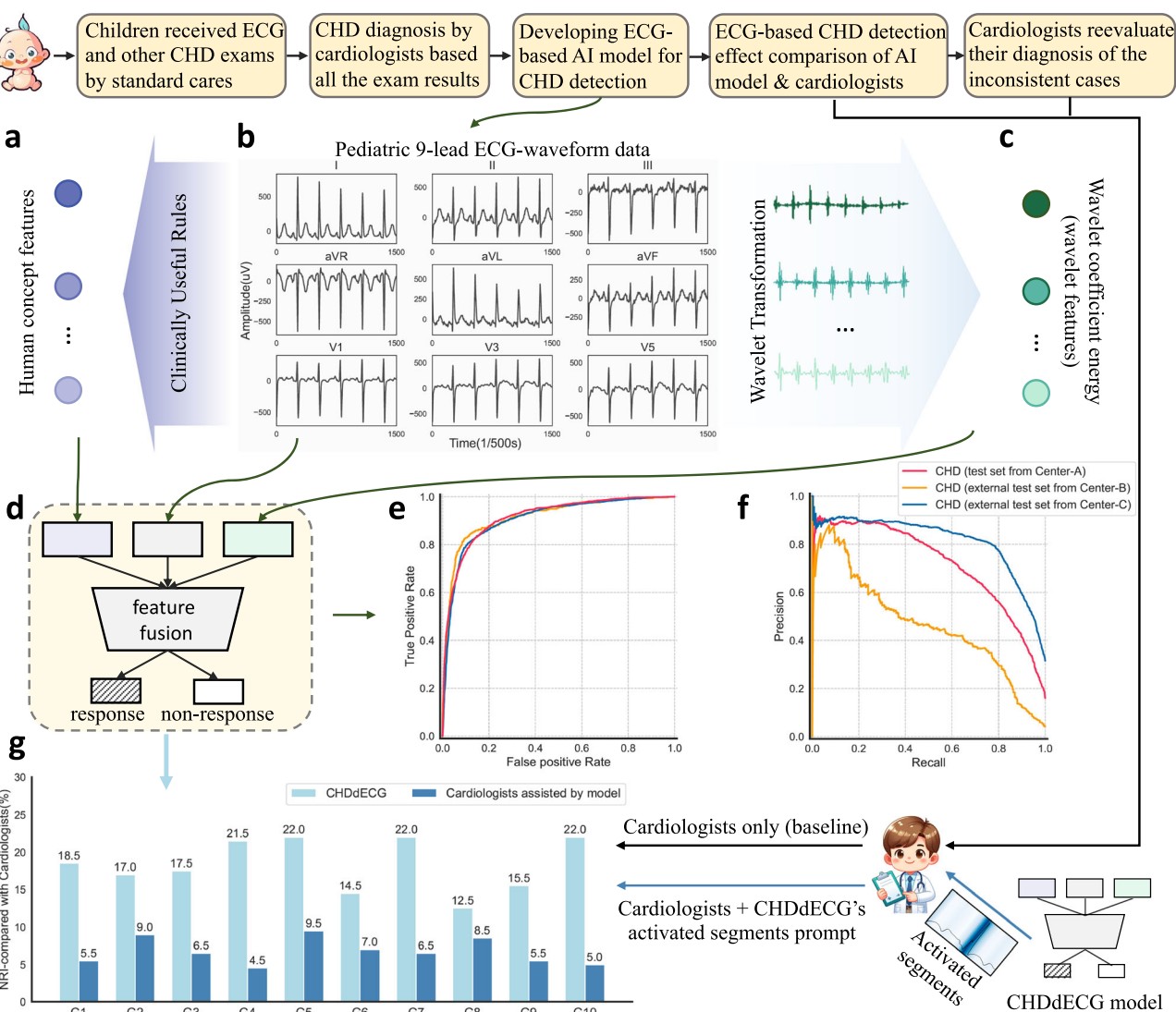

**Fig. 1 | The workflow of AI-enabled CHD detection with ECG data.** Hand-crafted human-concept features (**a**) were computed with some rules (corresponding formulas were in the Supplementary Materials) on pediatric ECG-waveform data (**b**), while the wavelet coefficient energy characteristics (as wavelet features (**c**)) obtained by performing wavelet transformation on the pediatric ECG-waveform data. Features of these three types were fed into the proposed AI model (**d**) for automatic fusion and CHD detection. **e** and **f** Illustrated the receiver operator characteristic curves and precision-recall curves of the AI model's CHD detection performances on a test set and two external test sets. **g** Illustrated the CHD detection effects (by the net reclassification index (NRI)) of the AI model and cardiologists assisted by the AI model, compared with cardiologists without any assistance as a baseline, across 10 randomly sampled test data groups (from the Center-A test set). The analysis of NRI(+) and NRI(−) was included in the Supplementary Materials. Source data are provided as a Source Data file.

test set of 12,000 cases and two external test sets of 7137 and 8121 cases, respectively. We found the ECG-based CHD identification by CHDdECG was promising, and more accurate than ECG cardiologists. Finally, we analyzed the prediction mechanisms of the trained CHDdECG and evaluated its robustness and reliability. Through exploring the potentials of pediatric ECG for CHD diagnosis, our CHDdECG made advances in three folds: (1) predicting structural heart defects of pediatric patients using ECG data; (2) drawing potential knowledge from pediatric ECG data beyond the current knowledge of experts through a deep learning approach; and (3) providing clues for further studies of pediatric ECG and CHDs.

## Results

### The overall CHD detection effects of CHDdECG

We trained the CHDdECG model to process pediatric ECG data for predicting the presence or absence of CHDs, without distinguishing between CHD subtypes. After training CHDdECG from scratch in a supervised learning manner, we evaluated the performances of the trained model on the internal test set and two external test sets. See the CHD prediction performances (sub-types are not distinguished) in Table 1, both the specificity and sensitivity of CHD detection exceeded 0.8 on the internal test set. On the two external test sets, characterized by different subtype proportion distributions, the specificity values were 0.937 and 0.907, respectively, and the sensitivity scores approached 0.8. Additionally, the median values of the probabilistic predictions (refer to the last column of Table 1) exhibited proximity to 1.0 on the internal test set and -0.8 on both external test sets. This highlights the high confidence and robust generalizability of CHDdECG's predictions (see Fig. 1e), the areas under the AUC curves (ROC-AUC), a comprehensive metric, were quite high, at 0.915 on the internal test set and at 0.917 and 0.907 on the two external test sets. The Brier scores were close to 0.0 on all the test sets. Figure 1f shows the precision-recall (PR) curves. Since they were influenced by class imbalance, the Center-C external test set showed a better PR-AUC score compared to the internal test set. All these comprehensive metrics signified that CHDdECG has achieved good performances and robust generalizability in CHD detection.

### Pediatric ECG-based CHD prediction outcomes compared to ECG cardiologists

We compared the CHD diagnosis outcomes of the trained CHDdECG model to those of 10 senior ECG cardiologists, divided into 10 groups denoted as G1–G10 in Fig. 1g. In each group, we randomly selected 200 ECG data with CHD and 100 non-CHD ECG data from the internal test set, with non-overlapping data between groups. For each group, the CHDdECG model made probabilistic predictions of CHD and required the ECG cardiologists to identify the CHD cases. We compared the performance of each method by computing the net reclassification index (NRI). As shown by the light blue bars in Fig. 1g, the diagnosis outcomes of cardiologists are regarded as the baseline, and the NRI scores for CHDdECG were much >0, indicating its superiority in pediatric ECG-based CHD detection compared to cardiologists. Furthermore, we specifically picked out those cases that are misidentified by cardiologists but are correctly identified by CHDdECG, prompting a reevaluation by the cardiologists. To aid their reevaluation, we also included the corresponding highlighted key ECG segments, as demonstrated in Fig. 2. Based on the prompt from CHDdECG, cardiologists identified some wavelets associated with CHD and changed parts of their original diagnosis. The results are illustrated as the dark blue bars in Fig. 1g, which indicate that the reevaluation outcomes are consistently better than the initial diagnosis results. However, the NRI on cardiologists' reevaluation remained inferior to CHDdECG, suggesting that some cases are still indistinguishable for cardiologists.

### CHD-related manifestation detection performances for major subtypes

We tested whether CHDdECG could effectively detect abnormal manifestations of major CHD subtypes. The definitions of subtypes follow the 2020 ESC guideline[50]. We fine-tuned the trained CHDdECG model for specific CHD subtypes. The ROC-AUC (area under the receiver operator characteristic curve) scores obtained on these test cases are reported in Table 1, spanning a range of 0.835 to 0.992 on the internal test set, as well as 0.889−0.926 and 0.859−0.939 on the two external test sets, respectively. Among the three most common subtypes (i.e., the ventricular septal defect, atrial septal defect, and patent ductus arteriosus), the ROC-AUC scores were 0.920, 0.835, and 0.856 on the internal test set, while achieved 0.918, 0.926, and 0.889 on the external test set from Center-B, and 0.913, 0.916, and 0.904 on the external test set from Center-C. On the internal test set, performances on 9 of 12 subtypes achieved high ROC-AUC scores over 0.9; on the external test set from Center-C, ROC-AUC scores on 7 of 9 subtypes are over 0.9. It is obvious that CHDdECG performs effectively on most subtypes, except for the relatively lower sensitivity scores on some subtypes (e.g., atrial septal defect (ASD) and patent ductus arteriosus (PDA) on the internal test set). The sensitivity on PDA is also relatively lower on the two external test sets. Notably, all the Brier scores are close to 0.0 on any test cohorts.

### Feature importance observation

To evaluate the prediction mechanisms of CHDdECG, we assessed the interpretability of the trained models (for CHDs and their various subtypes). We computed the importance scores of the feature types used for CHD detection, including raw ECG-waveform data, wavelet features, and hand-crafted human-concept features. The feature importance scores of 300 randomly selected test CHD cases (from the internal test set) were illustrated in a heat map in Fig. 3b, and feature importance scores were represented in the instance-specific view in Fig. 3a and feature-wise view in Fig. 3c (see Fig. 3a), features automatically extracted from ECG-waveform data supplied more information for predicting CHD statuses in most cases. The global feature-wise importance scores also affirmed this, while the features representing some human concepts and wavelet features were much less important, yet still beneficial. Here we gave a concrete case in Fig. 3d for ease of understanding. In this case, the global importance score of clinical features was 0.102, and that of wavelet features was 0.014, while the automatically extracted features from waveform data attained a score of 0.884. In Fig. 3e over subtypes, it can also be seen that the automatically extracted features yielded higher importance scores than the other feature types. The comparative experiments on the impact of different feature types on performance were provided in the Supplementary Materials.

### Key pediatric ECG segments of particular interest

Since we have observed that CHDdECG automatically extracted some critical features, we kept on exploring what ECG manifestations were adopted by CHDdECG using the Grad-CAM approach[51], and visualized the salient segments contributing to the CHD predictions in Fig. 2. We obtained the salient segments with Guided-Backpropagation[52] following the procedure of Grad-CAM algorithm[51], on the Temporal Attention layer's output features (refer to Fig. 5). Interestingly, we found that CHDdECG-activated segments were partially consistent with the previous observations on adult ECG data. In Fig. 2, we marked CHDdECG-activated segments with blue, and the darker blue indicated the more important segments. Figure 2a represented a notch on the R wave of signals of lead II, which was a typical abnormal manifestation with the atrial septal defect[27]; Fig. 2b illustrates the Katz−Wachtel phenomenon representing diphasic RS complexes on signals of lead V3, which was found to be related to cases with the ventricular septal defect[25]; Fig. 2c illustrated a QRS complex with a small R wave and a deep S wave on

**Table 1 | Model performances on the internal test set from Center-A, an external test set from Center-B, and another external test set from Center-C**

| Internal Test Set from Center-A | | | | | | |
|---|---|---|---|---|---|---|
| Categories | Prop(%) | ROC-AUC | Spec | Sens | Brier | Probabilistic Predictions |
| Atrial septal defect | 12.2% | 0.835 | 0.910 | 0.504 | 0.0736 | |
| Patent ductus arteriosus | 6.7% | 0.856 | 0.912 | 0.522 | 0.0673 | |
| Anomalous origin of a coronary artery | 0.9% | 0.894 | 0.938 | 0.722 | 0.0459 | |
| Ventricular septal defect | 36.6% | 0.920 | 0.925 | 0.707 | 0.0648 | |
| Coarctation of the aorta | 1.9% | 0.935 | 0.935 | 0.763 | 0.0487 | |
| Total anomalous pulmonary venous connection | 0.8% | 0.944 | 0.946 | 0.706 | 0.0385 | |
| dextro-Transposition of the great arteries | 2.1% | 0.929 | 0.952 | 0.721 | 0.0349 | |
| Pulmonary atresia | 2.5% | 0.985 | 0.966 | 0.902 | 0.0255 | |
| Single ventricle | 0.9% | 0.985 | 0.959 | 0.947 | 0.0284 | |
| Tetralogy of fallot | 3.3% | 0.987 | 0.966 | 0.927 | 0.0256 | |
| Atrioventricular septal defect | 2.0% | 0.991 | 0.946 | 0.974 | 0.0372 | |
| Double-outlet right ventricle | 0.8% | 0.992 | 0.963 | 1.000 | 0.0267 | |
| CHD (sub-types not distinguished) | - | 0.915 | 0.881 | 0.800 | 0.0998 | |

| External Test Set from Center-B | | | | | | |
|---|---|---|---|---|---|---|
| Categories | Prop(%) | ROC-AUC | Spec | Sens | Brier | Probabilistic Predictions |
| Atrial septal defect | 28.3% | 0.926 | 0.958 | 0.694 | 0.0357 | |
| Patent ductus arteriosus | 12.0% | 0.889 | 0.961 | 0.500 | 0.0325 | |
| Ventricular septal defect | 33.3% | 0.918 | 0.969 | 0.670 | 0.0296 | |
| CHD (sub-types not distinguished) | - | 0.917 | 0.937 | 0.770 | 0.0568 | |

| External Test Set from Center-C | | | | | | |
|---|---|---|---|---|---|---|
| Categories | Prop(%) | ROC-AUC | Spec | Sens | Brier | Probabilistic Predictions |
| Atrial septal defect | 52.2% | 0.916 | 0.936 | 0.705 | 0.0825 | |
| Patent ductus arteriosus | 7.9% | 0.904 | 0.938 | 0.673 | 0.0538 | |
| Anomalous origin of a coronary artery | 0.6% | 0.876 | 0.960 | 0.533 | 0.0340 | |
| Ventricular septal defect | 23.4% | 0.913 | 0.948 | 0.632 | 0.0616 | |
| Coarctation of the aorta | 1.0% | 0.859 | 0.956 | 0.375 | 0.0365 | |
| Pulmonary atresia | 2.3% | 0.906 | 0.976 | 0.842 | 0.0221 | |
| Single ventricle | 0.6% | 0.929 | 0.973 | 0.733 | 0.0203 | |
| Tetralogy of fallot | 3.7% | 0.939 | 0.976 | 0.837 | 0.0216 | |
| Atrioventricular septal defect | 0.7% | 0.909 | 0.966 | 0.611 | 0.0279 | |
| CHD (sub-types not distinguished) | - | 0.907 | 0.907 | 0.786 | 0.1043 | |

The reported performances for detecting CHDs or non-CHD were obtained with the CHDdECG model trained from scratch, while the performances for various CHD subtypes were obtained separately after the CHDdECG model was specifically fine-tuned. Notably, some subtypes were not included since their amounts were <0.5% of all the CHD cases or <10. The probabilistic predictions are represented by box-plots for better viewing of the distributions. Box-plot elements are defined as center line, median; box limits, upper and lower quantiles; whiskers, 1.5 × interquartile range; $n$ = 2038 CHD cases, 300 CHD cases, and 2521 CHD cases from Center-A, Center-B, and Certer-C, respectively. Source data are provided as a Source Data file.

signals of lead II, which was a typical manifestation in the left precordial leads of the cases with the dextro-transposition of the great arteries[53]; Fig. 2d showed an ST-segment elevation on the lead II ECG signals, which often occurred in the cases with the tetralogy of fallot[53]. We circled the representative malformations with regard to CHDs in Fig. 2 (in orange color) following previous observations on adult ECG[25,27,53]. It is obvious that the blue portions are highly overlapped with the orange circles, implying that the CHD predictions of CHDdECG were made based on relevant ECG segments. Besides, we also presented heatmaps for two CHDdECG-misidentified cases in

Fig. 2e and f. Figure 2e displayed a segment of the ECG signal that CHDdECG failed to correctly identify as the waveform of a ventricular septal defect. Although it exhibits similarities to the Katz–Wachtel phenomenon shown in Fig. 2b, we observed that its amplitude is significantly smaller (maximum values around 1000 μV), signifying an atypical form[25] not prioritized by the model (indicated by the lighter blue color compared to the waveform in Fig. 2b). Regarding Fig. 2f, CHDdECG might misidentify the double-humped waveform as a notch, however, it's ground truth diagnostic label is non-CHD. It is evident from Fig. 2e and f that the waveforms associated with CHD (or a

successful cases:

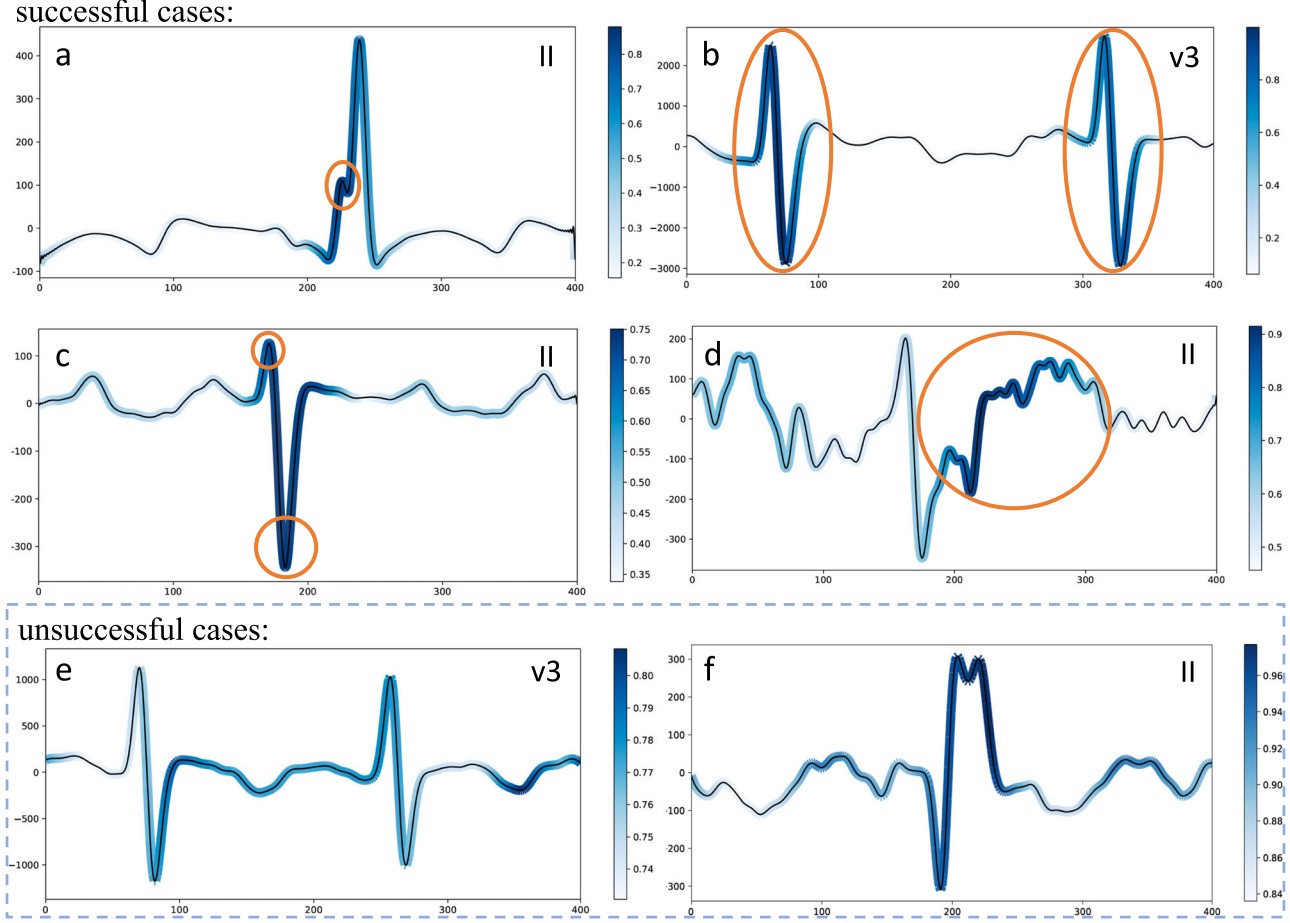

**Fig. 2 | Visualization of CHDdECG-activated segments for CHD subtypes using the Grad-GAM approach. a–d** Illustrated the classical ECG manifestations of subtype ASD, VSD, d-TGA, and TOF, respectively. **e** illustrated the ECG manifestations of subtype VSD but misidentified as non-CHD by CHDdECG. **f** illustrated a segment of non-CHD ECG manifestations but misidentified to be CHD by CHDdECG. The leads of ECG views are marked on the top right. The salient ECG segments were marked by blue (the darker blue indicated the more important segments). We further circled the typical manifestations in orange that were considered to be associated with subtypes of adult ECG by the cardiologists following previous research results[25]. The horizontal axis represents time ($4 \times 10^{-2}$ s), and the y-axis shows the amplitude of the electrocardiogram (μV). Source data are provided as a Source Data file.

specific subtype) are diverse and therefore challenging to detect. In summary, CHDdECG demonstrates the ability to identify certain congenital heart disease-related waveforms, and the visualization results are partially interpretable to humans, despite occasional misjudgments of confusing waveforms (which can also serve as a learning opportunity for humans).

## Discussion

Previous research indicated that the delayed detection of congenital heart disease (CHD) is a widespread issue across areas of varying income levels[8,9,21,54], which leads to missed opportunities for timely interventions. Besides, it is also recognized that the distribution of CHD subtypes can vary by location and over time[54]. In this study, we developed a pediatric-ECG-based differential CHD diagnosis approach, CHDdECG. CHDdECG was trained on large-scale real-world pediatric ECG data, and its effectiveness was validated on internal and external test sets, as presented in Table 1. The performance across comprehensive metrics, including specificity, sensitivity, ROC-AUC, and Brier scores, demonstrated the model's ability to accurately distinguish CHD-related ECG manifestations. Furthermore, the effectiveness of CHDdECG presented on two external test sets, characterized by different CHD subtype proportions and variations in ECG recording devices, suggested that the practical impact of subtype proportions

and device differences is limited in the application of CHDdECG (see Table 1 and Fig. 2), additional results indicated that CHDdECG also performed well in detecting specific manifestations of most CHD subtypes, suggesting CHDdECG's good generalization across varying subtypes and can be reliably used in practice.

Though the performances were generally good for most of the major CHD subtypes (see Table 1; especially for the tetralogy of fallot, atrioventricular septal defect, and double-outlet right ventricle), we also noticed that the detection sensitivity scores for some CHD subtypes (e.g., ASD and PDA on the internal test sets) were comparatively lower. Especially, the sensitivity scores for PDA were lower than other subtypes across all three test sets. We thought that these inferior sensitivity scores might be attributed to the inconspicuous CHD-related manifestations since it had been observed on adult ECG data[25] that some CHD cases were clinically silent. To further confirm this, we checked the sensitivity scores of cardiologists' analyses for ASD and PDA, which were only 0.306 and 0.434 respectively (with an overall sensitivity of around 0.6). It indicated that senior ECG cardiologists could not find abnormal manifestations in most of those cases from pediatric ECG as well. A positive aspect was that CHDdECG still outperformed cardiologists on ASD and PDA detection, and the relatively lower sensitivity for ASD and PDA did not hurt the benefits of CHDdECG in overall CHD detection. On the external test set from

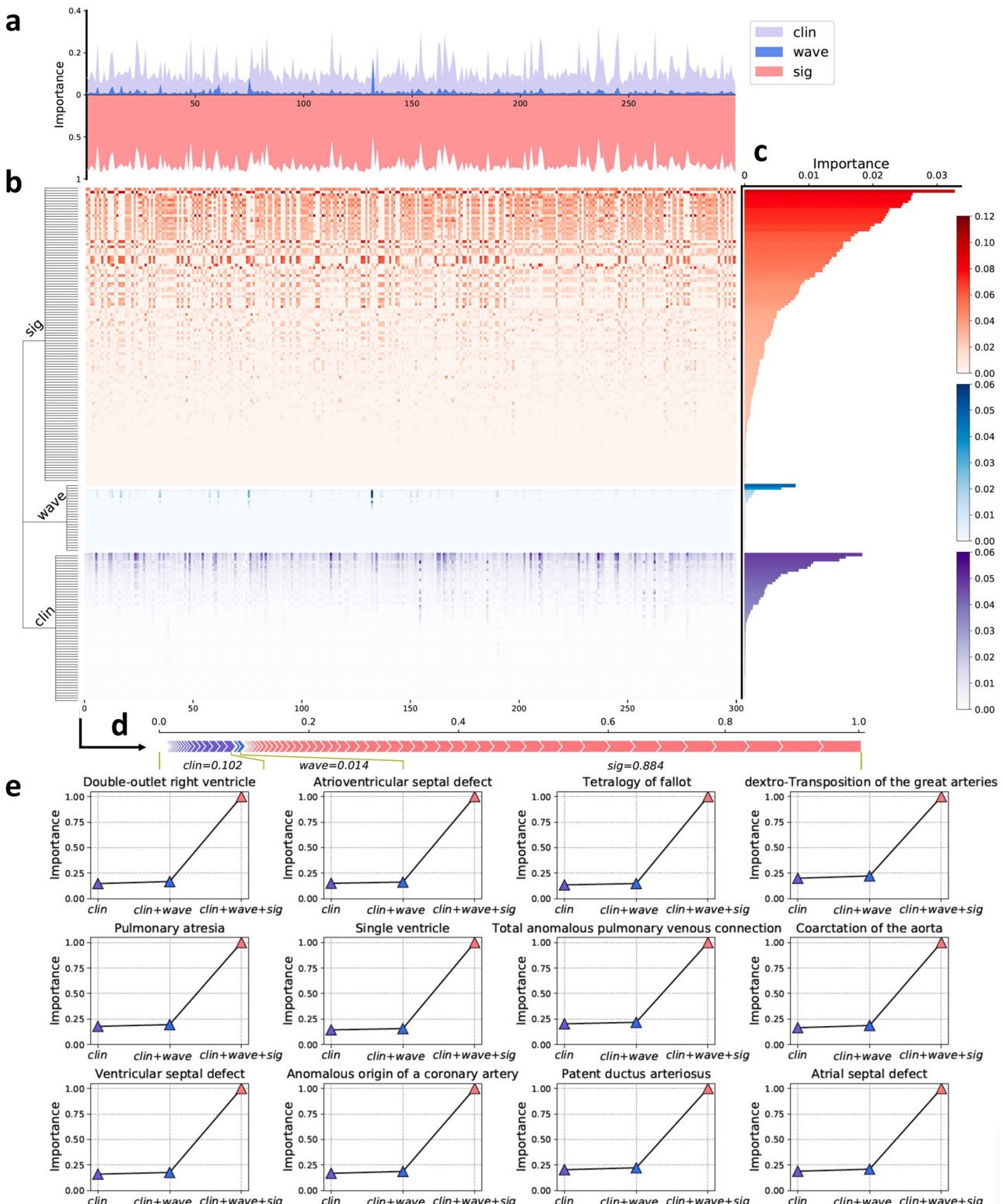

Fig. 3 | An illustration of feature importance of automatically extracted features from raw ECG signals (called sig), clinically useful human concept features (called clin), and wavelet features (called wave). We randomly selected 300 CHD cases from the test set and illustrated the feature importance scores with a heatmap (b) on these cases. The instance-wise feature importance scores (a) and the global importance scores (c) of features were also computed. d The feature importance scores of one case were especially shown. e An illustration of the feature importance of each feature type for various CHD subtypes (after the model fine-tuning). Note that the features with an importance score of zero were not included. Source data are provided as a Source Data file.

Center-C, we noticed the performances on the anomalous origin of a coronary artery (AOCA) and coarctation of the aorta (COA) were relatively lower. We further checked the sensitivity of cardiologists' analyses for these two subtypes, which were at 0.400 and 0.292, respectively, and were lower than those of CHDdECG. While CHDdECG's performances on Center-B appear notable in terms of ROC-AUC, specificity, and sensitivity, Fig. 1f uncovers a relatively low PR-AUC value of around 0.5. Conversely, the PR-AUC on Center-C showcases robust performance, surpassing 0.8. This phenomenon can be attributed to the significant label imbalance present in Center-B test set. In a nutshell, the CHD detection performances of CHDdECG implied the feasibility of using 9-lead pediatric ECG data to obtain differential diagnosis of CHD; but, the detection performances on some subtypes were sub-optimal due to the limited information in ECG signals.

Based on the robust performances of CHDdECG, we sought to shed light on the prediction mechanisms of CHDdECG. Adopting a deep learning approach to detect structural heart defects is theoretically based on an assumption that structural heart abnormalities can change the electrocardiovectors and thus lead to abnormal manifestations in ECG signals. However, some congenital cardiac malformations are subtle and do not show observable changes in the morphology of ECG signals. Hence, we have to examine whether CHDdECG's predictions were made based on reasonable features. See NRI compared to senior ECG cardiologists in Fig. 1g, we obtained three-fold findings: (1) CHDdECG is more effective in CHD detection than ECG cardiologists; (2) ECG cardiologists can achieve better CHD detection performances with the prompt of CHDdECG, which implied the predictions made by CHDdECG were reasonable and could be highly acceptable to experts; (3) ECG cardiologists cannot achieve CHDdECG-level performances even with the prompt of CHDdECG, suggesting that CHDdECG could extract some information out of human cognition. These results encouraged using the CHDdECG model for automatic CHD diagnosis since the prediction results were shown to be superior and highly trusty. It also encourages further studies to identify more hard-to-observe knowledge guided by CHDdECG.

Our further explorations attempted to enhance the clinical acceptance of the CHDdECG approach and facilitate interactions between cardiologists and CHDdECG by comparing the feature importance among the three feature types used by CHDdECG (i.e., automatically extracted ECG features, wavelet features, and concept features used by human experts). Analyses conducted from various perspectives (comparing the overall dataset level, subtype level, and instance level; see Fig. 3), all suggested that the automatically extracted features from ECG signals were the most important feature type and contributed more than the other two types, no matter for detecting the presence of CHD or for any specific subtypes. Furthermore, we observed that the key segments detected by CHDdECG (with Grad-CAM) presented many CHD-related malformations that could be observed in both pediatric ECG and adult ECG, despite that pediatric ECG is more complicated[55,56]. These findings implied that (1) the performance gains of CHDdECG (compared to ECG cardiologists; see Fig. 1g) might mainly come from the automatically extracted features, which represent some hard-to-observe information beyond the current human knowledge; (2) combining CHDdECG and a visualization approach (e.g., GradCAM) made such hard-to-observe knowledge much more accessible. The detailed analyses of pediatric ECG waveforms encouraged further investigations of the association between pediatric ECG and CHDs from theoretical and clinical perspectives.

One key strength of our study is that CHDdECG was devised to identify young CHD patients by using only routinely acquired pediatric ECG, thus enabling efficient CHD detection and timely interventions. In this context, it is more clinically meaningful than the previous studies on adult CHD cases. Note that we did not intend to replace the standard CHD diagnosis guideline for pediatric ECG. However, since there are economically underprivileged populations that have much less access to modern technologies and suffer from delayed interventions, we argue that, in these situations, it is highly desired to detect CHD in young children using our CHDdECG with pediatric ECG data, because it is reliable, low-cost, highly efficient, and has been verified on large-scale real-world datasets. Another key strength of our study is that the superior performances (e.g., outperforming ECG cardiologists) of CHDdECG can provide some potential knowledge on pediatric ECG beyond the current human knowledge. Thus, CHDdECG can offer clues for further exploring the potential of pediatric ECG data, which is generally beneficial.

There are still a few limitations in this study. First, although the test data we used aimed to follow real-world scenarios and we also collected external test data to examine the generalization capability of CHDdECG, the data distribution of CHDs can vary in different areas and times[54], which may cause somewhat different effects of CHDdECG, especially in situations when the subtype proportions are different from our test set and external test sets. The geographic specificity and fixed period of training and validation limit the assessment of generalizability. Second, 9-lead ECG data provide less information than the standard 12-lead ECG. Since putting more leads on the chests of young children was generally quite intractable, we made a trade-off decision to train CHDdECG on 9-lead pediatric ECG for wider application scenarios. Nevertheless, CHDdECG allows the processing of ECG data of any lead count, and we believe that the performances of CHDdECG will be better if it is trained and evaluated on standard 12-lead ECG data and CHDdECG can serve adults and elder children well. Third, our CHDdECG architecture is of high compatibility which allows the automated processing and fusion of multiple feature types. However, we used only three feature types for CHD detection, and some other feature types (e.g., signal features extracted by Bayesian approaches) might be further beneficial if they are included in consideration. Fourth, while our retrospective study has demonstrated the efficiency of CHDdECG on a real-world clinical dataset, the performance of CHDdECG for CHD screening in the general population remains uncertain, as it is challenging to prospectively obtain ECG data from children who do not necessarily require such examinations. Fifth, although the CHD labels were acquired following standardized diagnostic guidelines, we cannot rule out the possibility of label misclassification as a limitation, particularly when CHD cases present abnormalities below the level of human detection. This limitation also highlights the need for prospective protocol research dictating a comprehensive and standard diagnostic workup for all individuals.

## Methods

### Data access and ethical statement
This study was approved by the Medical Ethics Committee of Guangdong Provincial People's Hospital (KY-Q-2022-144-01). In accordance with ethical guidelines, this study secured a waiver for informed consent based on its retrospective analysis of anonymized data, ensuring privacy and security without explicit consent from subjects.

### Data sources
In this study, three distinct datasets were collected and used. The first dataset, utilized comprehensively for model training, validation, and internal testing, originates from the ECG Division in the Cardiovascular Outpatient Department at Guangdong Provincial People's Hospital (referred to as Center-A). The second dataset consists of an external test set sourced from the ECG Division in the Cardiovascular Inpatient Department at the same hospital (referred to as Center-B). Another external test set was obtained from the ECG Division at Shengjing Hospital of China Medical University (referred to as Center-C). These

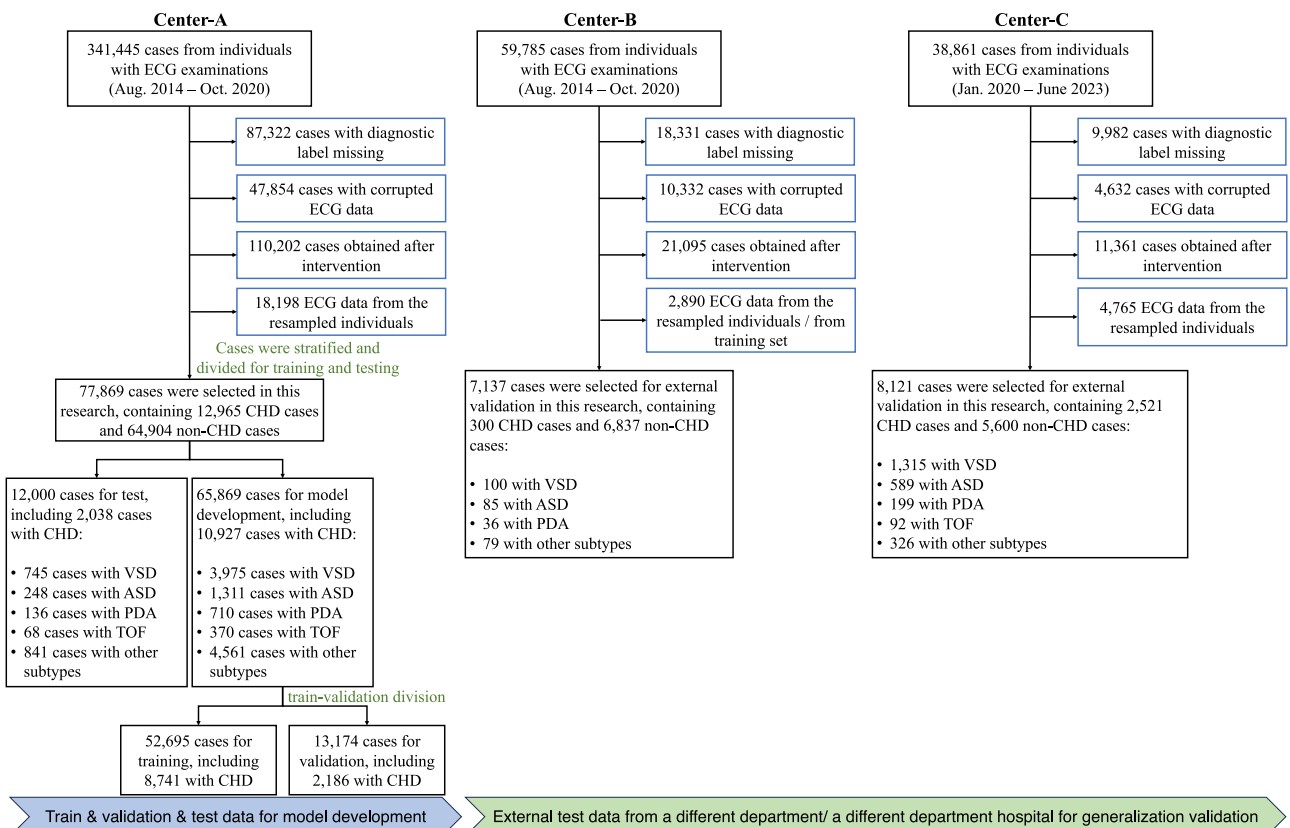

**Fig. 4 | An overview of the case selection procedures for Center-A, Center-B, and Center-C.** Center-A: the ECG Division in the Cardiovascular Outpatient Department at Guangdong Provincial People's Hospital; Center-B: the ECG Division in the Cardiovascular Inpatient Department at Guangdong Provincial People's Hospital; Center-C: the ECG Division at Shengjing Hospital of China Medical University. The descriptions within the blue boxes provided the reasons for omitting some cases (with ECG data and diagnostic results).

datasets collectively facilitated the development and comprehensive evaluation of the model. The ECG data in Center-A and Center-B were collected using identical ECG devices (GE MAC800) from August 2014 to October 2020, while the data in Center-C were collected utilizing a distinct brand of ECG device (NIHON KOHDEN ECG-2550) from January 2020 to June 2023. The data selection and train-validation-test split are illustrated in Fig. 4. For the sake of reaching reliable conclusions, some cases were omitted due to: (i) diagnosis results (labels) were missing; (ii) ECG signals were not correctly recorded, with excessive noises (over 20% signal amplitude values exceeding 5 mV), or with corrupted signals (20% recorded values are 0's); (iii) ECG cases were obtained after the intervention; (iv) ECG cases were obtained from the individuals whose other ECG-waveform data have been included in this study. The demographic and clinical characteristics of cohorts were reported in the Supplementary Materials. To ensure that patients from the test data sets were not included in the training data set, we excluded patients from the Center-B external test set if they were already present in the Center-A training set by using the enterprise master patient index (EMPI) which includes factors such as age, date of birth, sex, and name. For the Center-C external test set, a distance of almost 2300 km combined with comparison rules based on patients' name, sex, age, and CHD sub-type was used to ensure the absence of patient overlap with other centers. After that, 77,869 pediatric ECG cases from Center-A, 7137 cases from Center-B, and 8121 cases from Center-C finally remained for our study. Specifically, 65,869 cases (stratified around 85% of the ECG cases with various CHD subtypes or non-CHD ones; comprising 23,873 females and 41,996 males) in Center-A were randomly selected for model training, and the rest 12,000 cases (consisting of 4242 females and 7758 males) for model test. Notably, in this work, the sex of each participant was determined

based on their biological sex, as recorded on their Chinese identity card. In addition, the 7137 cases (consisting of 3458 females and 3679 males) in Center-B and the 8121 cases (3723 females and 4398 males) in Center-C comprised two independent external test sets, for evaluating the generalization of our CHDdECG. The CHDdECG model was trained to conduct CHD detection as a classification task in a supervised manner, and the classification labels used in the training phase indicating the CHD subtypes or the non-CHD status were real-world diagnostic results following standard diagnostic guidelines (e.g., using echocardiography) organized according to the International Statistical Classification of Diseases 10 codes (ICD-10).

All of the ECG cases for model training that we used were collected from individuals at the age of 2.12 ± 1.50 (year), among which the cases with CHDs were at the age of 1.58 ± 1.28 (year). This age distribution of our datasets satisfied the need to explore CHD detection methods for early intervention. Notably, over 90% of the cases had 9-lead ECG data with three missing chest leads, $V_2$, $V_4$, and $V_6$, because it was usually intractable to put all 6 chest leads on such a young child's chest. Thus, we built our framework based on 9-lead pediatric ECG data consisting of I, II, III, aVR, aVL, aVF, $V_1$, $V_3$, and $V_5$, and this setting would be easier to generalize in the young population. All pediatric ECG data were acquired at a frequency of 500 Hz over 10 s, and 5000 values on sampling points were obtained. As shown in Fig. 4, the CHD cases made up approximately 16.6% of Center-A dataset (8741 of 52,695 training cases, 2186 of 13,174 validation cases, and 2038 of 12,000 test cases), and approximately 4.2% and 26.12% of the Center-B and Center-C external test sets. The majority of the CHD cases belonged to the CHD subtypes of the ventricular septal defect (VSD), atrial septal defect (ASD), patent ductus arteriosus (PDA), and tetralogy of fallot (TOF), which is aligned with the real-world scenarios.

The quantitative proportions of the CHD subtypes are shown in the second column (Prop (%)) in Table 1. Given that the pediatric ECG data were sourced from diverse departments and hospitals without excessive selection, our collected datasets closely mirrored real-world medical scenarios and ensured our study was credible.

## Data pre-processing

Previous research suggested that some proper pre-processing on ECG-waveform data could lead to considerable performance gains[57]. Inspired by the successes of multi-modal data fusion approaches, we developed a CHDdECG model with three input branches, which took three types of features as input, including the ECG-waveform data $X_e$, the hand-crafted human-concept features $X_c$, and the features $X_w$ obtained by wavelet transformation. The last two types of features, $X_w$ and $X_c$, were organized in tabular data format. The inputs of the three branches were individually prepared as follows.

- First of all, we eliminated the noisy myoelectric signals (typically at 30–300 Hz) from the raw ECG-waveform data using the low-pass Butterworth filters[58,59]. Then, the interference of the electric power facilities (typically at 50 Hz) was eliminated by a finite impulse response notch filter[59] with the Kaiser window function[60]. Finally, the baseline wandering elimination was performed using the infinite impulse response zero-phase shift digital filter[59]. After these de-noising procedures, the key information of the ECG-waveform data was well preserved and the noise was partially eliminated. Then, we organized the ECG-waveform data into the format as $X_e \in \mathbb{R}^{9 \times 5000}$ (with 9 leads and 5000 sampling points in each lead).
- The wavelet features organized in a tabular data format, $X_w \in \mathbb{R}^{54}$, were obtained by performing the wavelet decomposition on the de-noised ECG signal $X_e$. We performed 9 levels of the wavelet decomposition with the db5 wavelet function, and the resulting coefficient energy characteristics of the 4th–8th levels were selected and concatenated into a feature vector (i.e., $X_w$). Note that in $X_w$, the elements were considered independent scalar features.
- The input human-concept features were also organized in a feature vector (i.e., $X_c \in \mathbb{R}^{114}$), whose elements were independent scalar features obtained from $X_e$. The scalar features in $X_c$ represent human concepts widely used in clinical ECG analysis. To imitate the clinical procedure to analyze ECG data, we first detected five keypoints (the P, Q, R, S, and T waves) on the axis using the findpeaks method of the Matlab Software. Specifically, to detect the inverted P and T waves, we took the absolute values of the sampling points on ECGs before using the findpeaks method. Then, the onset and end points of a peak (e.g., the R wave) were obtained by computing the slopes following the approach as in the literature[61]. After obtaining the keypoints on the axis, 114 tabular features were computed following the method[62] to provide clinically useful concepts, including the heartbeat rate, mean duration of QRS/P/PR segments, the mean amplitudes of P waves, et al. All of the formulas for computing the 114 scalar features were provided in the Supplementary Materials.

## Data normalization

After the pre-processing, $X_e$, $X_w$, and $X_c$ were respectively normalized with z-score (as in Eq. (1)) before being fed separately to the three input branches of the CHDdECG model (see Fig. 5), by

$$X'_i = \frac{X_i - \mu_i}{\sigma_i}, \tag{1}$$

where $X_i \in \{X_e, X_c, X_w\}$, $X'_i$ is the normalized outcome with the identical feature size, and $\mu_i$ and $\sigma_i$ are the mean and standard deviation of the $i$th component computed over the training set. For $X_e \in \mathbb{R}^{9 \times 5000}$, the normalization is performed along the lead dimension (i.e., the first dimension).

## CHDdECG architecture and data processing procedure

We proposed a deep learning-based model, CHDdECG, to use 9-lead pediatric ECG data for CHD detection. The model was implemented using the Keras framework[63] with Tensorflow 2.0 as the backend. CHDdECG mainly consisted of three input branches for three feature types and one output branch to make the probabilistic presence prediction for CHD. The input $X_e$ was sequentially processed by 1D convolution blocks, a three-path module consisting of 1D residual blocks[64] with various kernel sizes, a Transformer Encoder[65], and a temporal attention layer, to extract features in the local and global scopes. The features presented in tabular formats denoted as $X_w$ and $X_c$, were processed individually by TabBlocks[66]. We considered all the extracted features as independent scalar features and used one TabBlock[66] to select and fuse these features. Finally, the fused features were leveraged to predict the presence probability of CHD. The overall architecture of CHDdECG is shown in Fig. 5, and the detailed operations are depicted as follows.

1. For the ECG-waveform data $X_e$, we employed one-dimensional (1D) convolutional layers as the adaptive signal filters to extract the local signal features, regarding the 1D ECG signal as a special case of a 2D image. We first utilized two 1D convolutional layers (followed by batch normalization and a ReLU activation) to extract features along the temporal dimension. Then, three model paths were used, each of which contained three 1D residual blocks whose filter kernel sizes were, respectively, 3, 5, and 7. In this design, each 1D residual block down-sampled the features by 4 times along the temporal dimension. Notably, we used average pooling in the shortcut path of the residual blocks following ResNet-D[67]. The output features of the three model paths were concatenated along the channel dimension (since the features were organized into an identical size) and then fed to the Transformer Encoder module.

2. In addition to the convolutions used to extract local features, we also employed a Transformer Encoder block[65] to extract global features throughout the duration of ECG recording. The key component of the Transformer Encoder was a multi-head self-attention operation, which was defined by

$$h_i = \text{softmax}\left(\frac{(W_{Q,i}x)(W_{K,i}x)^T}{\sqrt{d_h}}\right)(W_{V,i}x), \qquad x_o = [h_1, h_2, \ldots, h_i, \ldots, h_n]W_o, \tag{2}$$

where $W_{Q,i}, W_{K,i}, W_{V,i} \in \mathbb{R}^{d_h \times d_x}$ and $W_o \in \mathbb{R}^{nd_h \times d_o}$ are learnable parametric matrices, $[\cdot, \cdot]$ denotes the concatenation operation, $d_x$ is the length of the input feature vector, $d_h$ is the hidden state dimension of the Transformer Encoder, $i$ is an index of the attention heads, $n$ is the number of the attention heads (in this study, we set $n = 8$), $x$ denotes the input feature, and $x_o$ denotes the output feature. After being processed by the self-attention module, the features were further processed by a feed-forward module composed of two linear layers with a ReLU activation in between (see Fig. 5).

3. Applying the convolutional operations and Transformer Encoder, the local features and global features were hierarchically extracted from the raw ECG-waveform data. After that, we employed a temporal attention layer to highlight the key segments, using a 1D convolution layer (with a batch normalization and a sigmoid activation) to compute spatial attention (see the right part of Fig. 5), as:

$$z_e = \text{Sigmoid}(\text{BatchNorm}(\text{Conv}(x_o))) \odot x_o, \tag{3}$$

where $\odot$ denotes point-wise multiplication, and $z_e$ is the output features of the temporal attention module.

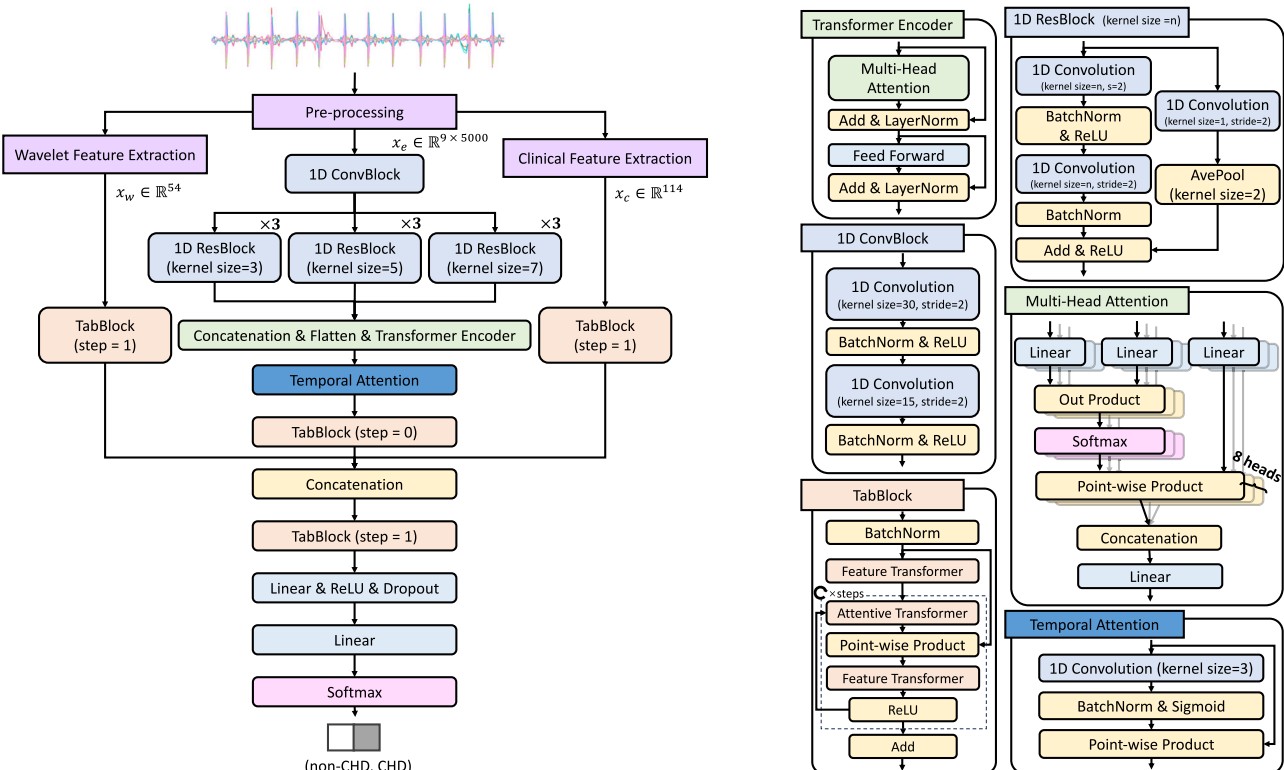

**Fig. 5 | An illustration of our proposed deep learning-based model, CHDdECG.** The left part showcases the overall architecture of the CHDdECG model, characterized by a fusion procedure involving multiple feature types. The right part presents the module details within CHDdECG. Please refer to the original TabNet paper[66] for the structure of the Attentive Transformer module and Feature Transformer module.

4. Finally, we treated the elements of the features $z_e$, $X_w$, and $X_c$ as independent scalar tabular features, and employed several Tab-Blocks to process them. For $z_e$, we flattened it into a feature vector before using the first TabBlock. A TabBlock contained an Attentive Transformer module for feature selection and a Feature Transformer for feature processing. Please refer to the original TabNet paper[66] for the detailed structure of the Attentive Transformer module and Feature Transformer module. The Attentive Transformer module computes a mask $m$ for feature selection, which filters out parts of input features by a point-wise multiplication. The selected scalar features were then processed by the Feature Transformer module within the TabBlock.

5. After the top-most TabBlock processing, we hypothesized that the higher-level semantic features from $X_e$, $X_w$, or $X_c$ associated with CHDs were effectively extracted and fused. These features were processed by two full connection layers with a BatchNorm layer and a ReLU activation in between, which were used to predict the presence and absence probabilities of CHD.

CHDdECG was trained in an end-to-end manner to jointly process the three types of input $X_e$, $X_w$, and $X_c$. Since there were relatively fewer cases with CHDs (compared to the non-CHD cases), we employed the label smoothing approach for the target $y$ in the training phase to avoid over-fitting, which is defined by:

$$\tilde{y} = \alpha y + (1 - \alpha)(1 - y), \tag{4}$$

where $\alpha \in [0, 0.5)$ is a hyperparameter coefficient, and $\alpha = 0.15$ was used in our study. In Eq. (4), the raw label $y$ ($y \in \{0, 1\}$) was obtained following the CHD diagnostic result ($y = 1$ if and only if the case was with CHD), and $\tilde{y}$ is the smoothed label used as the training target (obtained by Eq. (4)). The CHDdECG model was trained under the specification of the weighted cross-entropy loss function $\mathcal{L}$, defined by

$$\mathscr{L}(p, q) = -\tilde{y} \log(p) - w \cdot (1 - \tilde{y}) \log(q), \tag{5}$$

where $p$ denotes the predicted probability of a case with CHDs, $q$ denotes the probability of a case without CHDs, and $p$ and $q$ are convexly combined with the sum equal to 1 due to the final softmax layer. To deal with the class imbalance issue, the class weight parameter $w$ in Eq. (5) was set to 0.2, to make the model pay more attention to the CHD cases. In the training phase, CHDdECG for CHD detection was first initialized by He's parameter initialization[68] and was trained by 20 epochs from scratch using the Adam optimizer[69] with the default parameters. During training, the size of the mini-batches was 256. The learning rate was initialized to $1.0 \times 10^{-2}$ and was decayed by 10× every 8 epochs. In the validation and testing phases, CHDdECG inferred the CHD probabilities for the input ECG cases, using the parameters obtained in the training phase.

## Model fine-tuning for CHD subtype detection

To evaluate the capability of CHDdECG to detect the major CHD subtypes (each with a proportion over 0.5%), we fine-tuned the trained CHDdECG model to predict whether a case has characteristics of some CHD subtypes. Before the fine-tuning phase, we initialized the CHDdECG model with the parameters trained for overall CHD detection. During fine-tuning, we froze the parameters of the 1D ConvBlocks and the first 1D ResBlock in each sequential path and trained the other parameters for the target subtypes further with two epochs. In these fine-turning phases, we only used the target subtype cases and non-CHD cases. CHDdECG was fine-tuned under the guidance of Eq. (5) (with $w = 1$). Different from adopting the class-weighting strategy in training CHDdECG for CHD detection, we only employed the oversampling strategy to balance the probabilities of the usage of target subtypes and the non-CHD cases, since the sample amounts varied in different CHD subtypes.

## Importance score computing

Using TabBlocks also facilitated the computation of feature importance scores, following TabNet[66]. The Attentive Transformer module in the top-most TabBlock generates a data-specific sparse attention mask $m$, whose elements were in $[0, 1)$, as so to find useful features and to exclude useless features. The elements of $m$ could be interpreted as the importance of features. We denote $m_{n,i,j}$ as the importance score of the $j$th value in the heatmap for the $i$th feature type obtained using the $n$th ECG data. For better viewing, we computed the average importance scores of scalar features ($\bar{m}_{i,j}$), the overall importance score of the $i$th feature type, $\eta_i$, $i \in \{w, c, e\}$ (shown in Fig. 3e), and the feature type importance scores on each individual case, by

$$\bar{m}_{i,j} = \frac{\sum_{n=1}^{N} m_{n,i,j}}{N}, \eta_i = \sum_{j=1}^{n_i} \bar{m}_{i,j}, \bar{m}_{n,i} = \sum_{j=1}^{n_i} m_{n,i,j}, \quad (6)$$

where $N$ is the amount of ECG data, $n_i$ denotes the count of scalar features belonging to the $i$th feature type.

## Evaluation metrics

We comprehensively evaluated the prediction performance of CHD detection by employing several evaluation metrics. We employed the specificity, sensitivity, area under the receiver operating characteristic curve (ROC-AUC), Brier score, which were optimistic for imbalanced classification tasks. We also reported the probabilistic predictions by box plot. The definitions of these metrics were specified as follows:

- The sensitivity is a measure to evaluate how the model can predict the true positive cases, which is defined as

$$\text{sensitivity} = \frac{T_p}{T_p + F_n}, \quad (7)$$

where $T_p$ and $F_n$ denote the case amounts of true positives and false negatives, respectively.

- The specificity is a measure to evaluate how the model can predict the true negative cases, which is defined as

$$\text{specificity} = \frac{T_n}{T_n + F_p}, \quad (8)$$

where $T_n$ and $F_p$ denote the case amounts of true negatives and false positives, respectively.

- The Brier score is a strict measure to evaluate how good the probabilistic predictions are, which is defined by

$$\text{Brier score} = \frac{1}{N} \sum_{t=1}^{N} (f_t - o_t)^2, \quad (9)$$

where $T$ is the size of test set, $f_t$ is a probabilistic prediction and $o_t$ is the corresponding ground truth label.

- Since a higher sensitivity typically was with a lower specificity and vice versa, we also evaluated the performances of the ROC-AUC metric in Table 1. ROC-AUC is a graphical representation of the trade-off between a true positive rate and a false positive rate at various thresholds. It provides a comprehensive evaluation of the model performances.

- The probabilistic predictions were a statistic of the outcome probabilities yielded by the CHDdECG models for all the cases belonging to the target classes (CHD or some CHD subtypes). We displayed the probabilistic prediction outcomes by box plots in Table 1.

## Reporting summary

Further information on research design is available in the Nature Portfolio Reporting Summary linked to this article.

## Data availability

All data supporting the findings described in this manuscript are available in the article and in the Supplementary Information or/and from the corresponding author upon request. The ECG data used in this study cannot be shared publicly due to privacy restrictions. However, in the case of non-commercial use, researchers can sign the Data Access Form and Data License provided at the Github repository (https://github.com/shuaih720/CHDdECG) and contact H. Liang (lianghuiying-g@hotmail.com) to access the de-identified representative ECG data. Generally, we will respond within one week. Access will be granted by the data access committee. We have also deposited some representative data at the Github repository (https://github.com/shuaih720/CHDdECG), which is publicly available for scientific research and non-commercial use. Source data used to generate the tables and figures are provided with this paper. Source data are provided with this paper.

## Code availability

Our codes are available at GitHub: https://github.com/shuaih720/CHDdECG and Zenodo: https://doi.org/10.5281/zenodo.10477578.

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

## Acknowledgements

This study was funded by the National Key Research and Development Program of China (2019YFB1404803 to H.L., 2019YFC0118802 to J.W.), General Program of National Natural Science Foundation of China (62176231 to J.W. and 62076076 to H.L.), Excellent Young Scientists Fund (82122036 to H.L.), Zhejiang public Welfare Technology Research Project (LGF20F020013 to J.W.), and Young Scientists Fund of National Natural Science Foundation of China (82200558 to D.L.).

## Author contributions

J.C., S.H., Q.C., Y.Z. (Ying Zhang), D.L., Y.Z. (Yixiao Zhang), J.Q., L.H., X.P, Y.D., Y.G., D.C., J.W., and H.L. collected and analyzed the data. H.L., J.W., and A. B. conceived the project. H.L., J.W., A.B., J.C., and S.H. wrote the manuscript. All authors discussed the results and reviewed the manuscript.

## Competing interests

The authors declare no competing interests.

## Additional information

¹State Key Laboratory of Transvascular Implantation Devices of the Second Affiliated Hospital, Zhejiang University School of Medicine, Zhejiang University, 310009 Hangzhou, China. ²Medical Big Data Center, Guangdong Provincial People's Hospital (Guangdong Academy of Medical Sciences), Southern Medical University, 510080 Guangzhou, Guangdong Province, China. ³Guangdong Cardiovascular Institute, Guangdong Provincial People's Hospital, Guangdong Academy of Medical Sciences, 510080 Guangzhou, Guangdong Province, China. ⁴Guangdong Provincial Key Laboratory of Artificial Intelligence in Medical Image Analysis and Application, Guangdong Provincial People's Hospital (Guangdong Academy of Medical Sciences), 510080 Guangzhou, Guangdong Province, China. ⁵Department of Cardiology, Guangdong Provincial People's Hospital (Guangdong Academy of Medical Sciences), Southern Medical University, 510080 Guangzhou, Guangdong Province, China. ⁶Liaoning Engineering Research Center of Intelligent Diagnosis and Treatment Ecosystem, 110004 Shenyang, Liaoning Province, China. ⁷Clinical Research Center of Shengjing Hospital of China Medical University, 110004 Shenyang, Liaoning Province, China. ⁸Department of Urology Surgery, Shengjing Hospital of China Medical University, 110004 Shenyang, Liaoning Province, China. ⁹College of Information Technology and Engineering, Guangzhou College of Commerce, 510363 Guangzhou, Guangdong Province, China. ¹⁰Guangzhou Women and Children's Medical Center, Guangzhou Medical University, 510623 Guangzhou, Guangdong Province, China. ¹¹Zhuhai Precision Medical Center, Zhuhai People's Hospital/ Zhuhai Hospital Affiliated with Jinan University, Jinan University, 519000 Zhuhai, Guangdong Province, China. ¹²The Biomedical Translational Research Institute, Jinan University Faculty of Medical Science, Jinan University, 510632 Guangzhou, Guangdong Province, China. ¹³Department of Computer Science and Engineering, University of Notre Dame, Notre Dame, IN 46556, USA. ¹⁴Institute of Sciences in Emergency Medicine, Guangdong Provincial People's Hospital, Guangdong Academy of Medical Sciences, 510080 Guangzhou, China. ¹⁵Department of Emergency Medicine, Wayne State University School of Medicine, Detroit, MI 48201, USA. ¹⁶School of Public Health, Zhejiang University, 310058 Hangzhou, China. ¹⁷These authors contributed equally: Jintai Chen, Shuai Huang, Ying Zhang, Qing Chang. ✉e-mail: abellou402@gmail.com; wujian2000@zju.edu.cn; lianghuiying@hotmail.com

