## [Peer Review File · Nature Communications]

Reviewers' Comments:

Reviewer #1:

Remarks to the Author:

In the current study, the authors develop a deep learning approach to the detection of congenital heart disease using pediatric 9-lead ECGs. The model fuses three types of data: ECG waveform data, manually specified ECG features, and wavelet transformation. The model is trained in a large tertiary referral center and validated in an internal test set and an external test set (a second large tertiary referral center). The model demonstrates good discrimination performance, with apparent higher sensitivity for CHD when compared to cardiologist readers. Model interpretation techniques suggest that ECG signals contribute the majority of predictive information, and saliency maps suggest that plausible regions of the ECG appear to underlie model performance.

The work appears largely sound, represents an extension of prior literature (deep learning models have generally not focused on CHD), and there is clinical relevance here. There are substantial limitations however, which predominantly relate to sampling, disease definition, and QC methods which may introduce bias. Model evaluation is also lacking in some components. Specific comments follow:

1. The Introduction contains the necessary setup but is long, verbose and difficult to follow. The general concept is that there is reason to believe that the ECG may encode risk for occult CHD in children, and that a deep learning model may be able to extract it efficiently to enable early detection. I recommend revision to cut down on unnecessary words to increase clarity.
2. Clarification of phenotyping is needed. Specifically, reference is made to CHD detection "following the standard guideline with various tools", but no reference or guidelines are provided, and the description is vague. A clearer definition of how each CHD was defined would be useful.
3. Related to comment #2, all children in the study underwent ECG. The authors should provide further comment on the indications for this ECG – I assume that an ECG is not performed routinely (i.e., among patients in whom CHD is not suspected), and therefore the presence of an ECG for clinical purposes is subject to indication bias. If so, this is a substantial limitation and should be acknowledged, since it becomes unclear how the model would perform when applied prospectively among individuals who would not necessarily otherwise get an ECG (such as those who could be 'screened' for CHD as the authors suggest as a primary use case).
4. The presentation of the Results requires clarification. It is unclear what is meant by 'the medians of the prediction probabilities were quite significant', and the importance of this observation.
5. Related to comment #4 above, is there any assessment of model calibration? How accurate are the predicted probabilities on an absolute basis? This is a fundamental component of model evaluation that appears lacking.
6. The authors argue that the deep learning model is more sensitive to CHD diagnosis than the cardiologists. This is only showing one side of the story (i.e., does the increase in sensitivity come at a cost of lower specificity?), and is therefore incomplete. A standard NRI analysis would be more customary and balanced.
7. The authors should consider displaying precision-recall curves in addition to ROC.
8. The model interpretation work is useful. However, I am having some trouble following Figure 3. I understand the saliency depiction, but how is the displayed tracing generated? Which lead(s) are represented? The cited 'signs' (e.g., Katz-Wachtel phenomenon) are specific to certain leads, so it is unclear whether the observations seen on the saliency map are truly consistent with expectations.
9. Clinically I find it a little counter-intuitive that the model performed so well for "all CHD" considering how varied the CHD entities (and their ECG signatures are). To me it raises some concern that the model may be picking up on some kind of bias factor, although the saliency maps do provide some reassurance against this. Could the authors comment?
10. The proportion of tracings excluded for 'corruption' is very high (40% in Ref A and 50% in Ref B). This is concerning, and may be a source of bias. More detail is needed regarding assessment of tracing quality and criteria for excluding tracings based on noise. Ideally an objective definition would be applied.
11. The authors argue that a potential value of the CHDdECG model is better detection of CHD in resource-poor settings, but the model was not evaluated in such settings (i.e., it was evaluated in two large tertiary care centers). I would also acknowledge geographic specificity of train/test as a potential limitation to generalizability.

Minor comments

- Frequent grammatical errors requiring correction
- Odd phrases that require revision, eg "highly trusty", "not excessively selected", "hand crafted"
- Can the authors confirm that the normalization parameters used for inference in both test sets represent those obtained from the training set?
- Was early stopping employed in training?

I have expertise in clinical cardiology, epidemiology and risk prediction, and clinical application of machine learning models. I cannot provide expert-level technical review of the deep learning model design.

Reviewer #2:

Remarks to the Author:

The reviewer has the following concerns:

- 1) The authors mentioned that they constructed two test sets, an independent test set and an external test set. Please clarify the data source of the first independent test set and the test set used in the comparison of sensitivity on detecting CHD-related malformations between CHDdECG and ECG cardiologists.
- 2) One thing worth noting is that the sensitivity scores of CHDdECG are generally higher than that of cardiologists among all the groups. And the score gaps are around 20%. This phenomenon may be caused by the biased training when the test set is collected from the same center as training set. Please add experiments to demonstrate the cases that the cardiologists accept the explainable evidence and change their decisions, especially on the samples misidentified by cardiologists but correctly identified by CHDdECG in the test set. These experiments will make the work more valuable and convincing.
- 3) In Figure 3, the authors showed explainable evidence for the four sub-types of CHD, and the authors selected specific heartbeats variant in morphology. To have an objective view on the explainable evidence, the authors should provide and discuss the gradient heatmap of the misidentified samples.
- 4) For data pre-processing, the authors normalized the three types of features using z-score. However, this approach will eliminate true amplitude for QRS-complex, which is the supportive evidence for some morphology anomaly. The authors should provide the reason for ignoring the amplitude information.
- 5) There are several confused issues for the hand-crafted features.
The first is about the "findpeaks" method from Matlab. This function will annotate all the peaks and can be easily disturbed by noise. The authors then used the method with slope in [61] to locate the onsets and offsets of the QRS-complex, P and T waves. In a technical perspective, the location should be erratic using this method, especially on P waves. Please validate the wave localization method on the open-access database with annotations for QRS-complex, P and T waves. The following database can be helpful.
LUDB database: <https://physionet.org/content/ludb/1.0.1/>
QT database: <https://physionet.org/content/qtdb/1.0.0/>
The second issue is that the authors should discuss why X_c is much less important than X_e while the inference stage of CHDdECG, since the ECG segmentation method is defective. More specifically, to verify if the insensitivity of CHDdECG on X_c is caused by inaccurate localization of ECG waves.
The third issue is that it is inappropriate to deem X_c as the features representing expert knowledge, since X_c is a common feature extraction based on ECG segmentation. It does not involve the discrimination of clinical information using expert knowledge. A more accurate description for X_c should be the features based on ECG segmentation. The authors can prove the capacity in knowledge discovery of CHDdECG with X_c and the explainable evidence. However, the method in this paper should not be defined as deep learning integrated with "Human Knowledge".
- 6) Please add ablation study for utilizing X_e , X_w and X_c as feature input separately. And the authors should discuss the effectiveness and rationality of using the three feature extraction methods according to the ablation results.

RESPONSE TO REVIEWERS

Of “Superior Detection of Congenital Heart Diseases by Pediatric Electrocardiogram Based Deep Learning Integrated with Human Knowledge (NCOMMS-22-45582-T)”

Dear reviewers,

We sincerely thank you for the insightful and constructive comments and giving us the opportunity to revise and promote the quality of this manuscript. Following your suggestions, we carefully revise the manuscript and try our best to reduce the typographical, grammatical, and bibliographical errors. Below is our point-by-point response to all the comments made by the reviewers. We hope our response will clarify all your concerns.

To Reviewer #1:

1. The Introduction contains the necessary setup but is long, verbose and difficult to follow. The general concept is that there is reason to believe that the ECG may encode risk for occult CHD in children, and that a deep learning model may be able to extract it efficiently to enable early detection. I recommend revision to cut down on unnecessary words to increase clarity.

Response: We appreciate and thank you for the suggestions. We agree that the previous version is too long, verbose and difficult to follow. We have removed some useless words, combined several sentences, changed the expressions to make the Introduction more concise and clear. Besides, we also modified the other parts of the manuscript to make the whole paper concise. The changed parts are marked in red. We hope our revised manuscript can clarify all your concerns.

2. Clarification of phenotyping is needed. Specifically, reference is made to CHD detection “following the standard guideline with various tools”, but no reference or guidelines are provided, and the description is vague. A clearer definition of how each CHD was defined would be useful.

Response: We would like to express our gratitude for your thoughtful comments. Please note that, at the tertiary referral centers of China, the doctors who diagnose congenital heart disease typically have excellent medical education backgrounds and sufficient clinical experience (this applies to our case). Therefore, although the specific diagnostic tests may vary slightly

depending on individual cases, their applications of the examination tools and the diagnosis of medical records are highly reliable and follow the guidelines in clinical practice.

Thus, in the revised manuscript, we clarify that:

“First, potential pediatric patients underwent several examinations, primarily consisting of transthoracic echocardiography (TTE) and electrocardiogram, in accordance with the European Society of Cardiology Guidelines for CHD. In certain cases, additional tests may have been used at the discretion of the attending doctors. The doctors carefully analyzed all the examination results and subsequently made the final diagnostic judgments.”

We also cited the guideline as follows:

“Baumgartner H, De Backer J, Babu-Narayan SV, Budts W, Chessa M, Diller GP, Lung B, Kluin J, Lang IM, Meijboom F, Moons P, Mulder BJM, Oechslin E, Roos-Hesselink JW, Schwerzmann M, Sondergaard L, Zeppenfeld K; ESC Scientific Document Group. 2020 ESC Guidelines for the management of adult congenital heart disease. Eur Heart J. 2021 Feb 11;42(6): 563-645.”

3. Related to comment #2, all children in the study underwent ECG. The authors should provide further comment on the indications for this ECG – I assume that an ECG is not performed routinely (i.e., among patients in whom CHD is not suspected), and therefore the presence of an ECG for clinical purposes is subject to indication bias. If so, this is a substantial limitation and should be acknowledged, since it becomes unclear how the model would perform when applied prospectively among individuals who would not necessarily otherwise get an ECG (such as those who could be ‘screened’ for CHD as the authors suggest as a primary use case).

Response: Thank you and we agree that it is still unclear how the model would perform when applied prospectively among individuals who would ****totally**** not necessarily otherwise get an ECG. However, it is very difficult to obtain the ECG of children who ****totally**** not necessarily to get an ECG.

Actually, the CHDdECG was validated on a dataset from a real-world scenario. Specifically, we utilized all the available pediatric electrocardiogram data from the referral centers, without any disease-relevant data selection. Many of used ECGs that were not issued by cardiology department for excluding surgical and medication risks (e.g., with the diseases of hydrocele of tunica vaginalis and oblique inguinal hernia). We believe that the inclusion of such data in our real-world dataset partially demonstrates the potential of the model for CHD screening. Besides,

since the CHDdECG was validated on real-world dataset, the benefits of using CHDdECG in CHD detection is still significant.

But, we agree with you that the statement of CHD screening is less rigorous, so we remove the statements about “CHD screening” and point out the disadvantage as (in the last paragraph of Discussion):

“While our retrospective study has demonstrated the efficiency of CHDdECG on a real-world clinical dataset, the performance of CHDdECG for CHD screening in the general population remains uncertain, as it is challenging to prospectively obtain ECG data from children who do not necessarily require such examinations.”

4. The presentation of the Results requires clarification. It is unclear what is meant by ‘the medians of the prediction probabilities were quite significant’, and the importance of this observation.

Response: Thank you for your suggestions. To clarify the statement, we have changed “prediction probabilities” to “probabilistic predictions”, which represent the estimated likelihood w.r.t. CHD (or subtypes). In Table 1, we used box-plots to display the distributions of probabilistic predictions, with the median values indicating the overall uncertainty of the model prediction. Median values close to 1 indicate that our CHDdECG model is confident in identifying CHD (or some certain subtype) cases. We have also included a definition of box-plot elements in the caption of Table 1, as follows: ‘center line, median; box limits, upper and lower quantiles; whiskers, $1.5\times$ interquartile range.’ We also reminded readers to refer to the last column of Table 1 when we first mention the “the median value of probabilistic predictions” in the main text for better understanding. We hope these revisions meet your expectations.

5. Related to comment #4 above, is there any assessment of model calibration? How accurate are the predicted probabilities on an absolute basis? This is a fundamental component of model evaluation that appears lacking.

Response: Thank you for your helpful suggestion. To assess the calibration of our CHDdECG model, we used the Brier score as a comprehensive measure, which we have added to Table 1 and explained in the Results section. It is obvious that all Brier scores obtained were close to 0.0, which indicates the trained CHDdECG is well-calibrated.

6. The authors argue that the deep learning model is more sensitive to CHD diagnosis than the cardiologists. This is only showing one side of the story (i.e., does the increase in

sensitivity come at a cost of lower specificity>), and is therefore incomplete. A standard NRI analysis would be more customary and balanced.

Response: Thank you for your insightful suggestion! We agree that comparing only sensitivity may not be sufficient to demonstrate the superiority of CHDdECG. As a result, we reorganized the comparison between cardiologists and CHDdECG in terms of CHD case identification, and the results are measured by NRI, rather than sensitivity. The comparison results are illustrated below.

We describe the experiments and the outcomes in the Result section, as follows.

“We compared the performance of the trained CHDdECG model to that of 10 senior ECG cardiologists, divided into 10 groups denoted as ‘G1’--‘G10’ in Fig.1 (g). In each group, we randomly selected 100 ECG data with CHD and 450 non-CHD ECG data from the test set, with non-overlapping data between groups. For each group, we used the CHDdECG model for probabilistic predictions of CHD and required the ECG cardiologist to identify the CHD cases. We assessed the performance of each method by computing the net reclassification index (NRI). As shown by the light blue bars in Fig.1 (g), the performances of cardiologists are regarded as the benchmark, and the NRI scores for CHDdECG were significantly greater than 0, indicating its superior performance in pediatric ECG-based CHD detection compared to cardiologists. Further, we picked out those cases (with the highlighted key ECG segments; see Fig.3 for example) misidentified by cardiologists but correctly identified by CHDdECG for cardiologists’ rechecking. Based on the prompt of CHDdECG, the dark blue bars in Fig.1(g) indicated that cardiologists change parts of their original judgements for the picked cases and thus improved their detection performances. However, the NRI of cardiologists’ rechecking remained inferior to CHDdECG, suggesting that some cases are still indistinguishable for cardiologists.”

7. The authors should consider displaying precision-recall curves in addition to ROC.

Response: Thank you for your suggestion. We have added the precision-recall curve representing the prediction performances for CHD cases to Fig. 1 of the revised manuscript (as shown in the Figure 1 below). However, we would like to gently point out that the precision-recall curve may not be the most suitable evaluation metric for extensive category-data-imbalanced classification, as it typically results in a low score in either precision or recall. Therefore, in such cases, the PR-AUC is typically used to compare model performances. To provide a more comprehensive performance comparison, we have included the PR-AUC values and ROC-AUC in the supplementary materials (see the Table 1 and Table 2 below).

Figure 1. Precision-recall curves on test set and external test sets.

Table 4. Classification performances by “ROC-AUC” on internal and external test sets from Ref-A and Ref-B.

Test Set from Ref-A						
	CHDdECG	CNN	LSTM	k -NN	RF	Xgboost
Atrial septal defect	0.835	0.793	0.815	0.721	0.804	0.819
Patent ductus arteriosus	0.856	0.799	0.835	0.756	0.789	0.803
Anomalous origin of a coronary artery	0.894	0.886	0.852	0.796	0.815	0.901
Ventricular septal defect	0.920	0.912	0.905	0.806	0.855	0.899
Coarctation of the aorta	0.935	0.915	0.930	0.816	0.861	0.923
Total anomalous pulmonary venous connection	0.944	0.920	0.925	0.831	0.886	0.930
dextro-Transposition of the great arteries	0.929	0.891	0.912	0.795	0.849	0.899
Pulmonary atresia	0.985	0.952	0.961	0.864	0.912	0.967
Single ventricle	0.985	0.960	0.966	0.897	0.932	0.990
Tetralogy of fallot	0.987	0.957	0.972	0.883	0.942	0.989
Atrioventricular septal defect	0.991	0.984	0.979	0.868	0.937	0.985
Double-outlet right ventricle	0.992	0.981	0.970	0.872	0.955	0.982
CHD (with all the sub-types)	0.915	0.875	0.882	0.777	0.851	0.906
External Test Set from Ref-B						
	CHDdECG	CNN	LSTM	k -NN	RF	Xgboost
Atrial septal defect	0.926	0.887	0.897	0.801	0.873	0.887
Patent ductus arteriosus	0.889	0.862	0.881	0.738	0.823	0.891
Ventricular septal defect	0.918	0.903	0.898	0.797	0.835	0.912
CHD (with all the sub-types)	0.917	0.868	0.873	0.754	0.861	0.889

Table 5. Classification performances measured by “PR-AUC” on internal and external test sets from Ref-A and Ref-B.

Test Set from Ref-A						
	CHDdECG	CNN	LSTM	k-NN	RF	Xgboost
Atrial septal defect	0.130	0.088	0.128	0.097	0.132	0.073
Patent ductus arteriosus	0.063	0.028	0.066	0.032	0.084	0.067
Anomalous origin of a coronary artery	0.027	0.020	0.025	0.027	0.020	0.011
Ventricular septal defect	0.540	0.362	0.346	0.141	0.291	0.395
Coarctation of the aorta	0.080	0.053	0.062	0.041	0.074	0.047
Total anomalous pulmonary venous connection	0.054	0.035	0.063	0.056	0.050	0.046
dextro-Transposition of the great arteries	0.096	0.038	0.071	0.065	0.043	0.023
Pulmonary atresia	0.233	0.136	0.148	0.113	0.157	0.133
Single ventricle	0.082	0.052	0.047	0.050	0.058	0.083
Tetralogy of fallot	0.361	0.223	0.241	0.014	0.320	0.242
Atrioventricular septal defect	0.241	0.152	0.175	0.096	0.173	0.049
Double-outlet right ventricle	0.160	0.166	0.085	0.071	0.066	0.102
CHD (with all the sub-types)	0.721	0.663	0.692	0.270	0.652	0.586
External Test Set from Ref-B						
	CHDdECG	CNN	LSTM	k-NN	RF	Xgboost
Atrial septal defect	0.264	0.278	0.218	0.091	0.221	0.221
Patent ductus arteriosus	0.114	0.092	0.094	0.041	0.144	0.152
Ventricular septal defect	0.288	0.256	0.197	0.079	0.202	0.266
CHD (with all the sub-types)	0.464	0.424	0.413	0.132	0.412	0.461

8. The model interpretation work is useful. However, I am having some trouble following Figure 3. I understand the saliency depiction, but how is the displayed tracing generated? Which lead(s) are represented? The cited ‘signs’ (e.g., Katz-Wachtel phenomenon) are specific to certain leads, so it is unclear whether the observations on seen on the saliency map are truly consistent with expectations.

Response: Thank you. The saliency depiction was generated using GradCAM (Gradient-weighted Class Activation Mapping), a widely-used visualization technique used to understand the features learned by models in making a particular classification decision. The technique generates a heat map to show the regions of the input data (e.g., image) that were important in making the classification decision. Grad-CAM works by computing the gradient of the output feature map of the models with respect to the target class. The resulting gradient is then used to calculate the importance of each activation in the feature map by averaging the gradients over the spatial dimensions. These importance scores are then weighted by the activation values in the feature map to generate the heat map. The saliency depiction is available because we treated the ECG signals similarly to images, where the 1D ECG signal can be regarded as a special case of a 2D image.

In addition, we would like to thank you for pointing out that we forgot to mark the lead of the visualized waves. We have added the lead information to the top right corners. We have also

visualized the signals of lead V3 in Fig. 3 (b), as the Katz-Wachtel phenomenon can occur in ECG signals of this lead.

9. Clinically I find it a little counter-intuitive that the model performed so well for “all CHD” considering how varied the CHD entities (and their ECG signatures are). To me it raises some concern that the model may be picking up on some kind of bias factor, although the saliency maps do provide some reassurance against this. Could the authors comment?

Response: Thank you for your attention to detail. We believe that there are three key reasons why AI outperforms individual doctors in CHD-related ECG analysis.

(1). Unlike languages or images, CHD-related ECG data are relatively rare, making it difficult for cardiologists to gain extensive experience. In contrast, our CHDdECG has been trained on a large amount of CHD-related ECG data, providing a learning experience that surpasses that of individual cardiologists.

(2). While specialized doctors’ high diagnostic accuracy for a disease is generally based on multiple examination results, the accuracy of diagnosing a disease based on a single examination (such as ECG) may not necessarily have an advantage.

(3). Some specific ECG changes are very subtle and difficult to identify visually but can be detected through AI or computer analysis. A CHD-irrelevant example is that, predicting sudden death with T-wave alternans is a specific ECG change that is difficult to distinguish visually but can be easily detected using specialized ECG analysis software. Although CHD-related ECGs are more complex, there may be similarly subtle yet meaningful ECG changes that are clinically helpful but unknown or invisible to humans.

10. The proportion of tracings excluded for ‘corruption’ is very high (40% in Ref A and 50% in Ref B). This is concerning, and may be a source of bias. More detail is needed regarding assessment of tracing quality and criteria for excluding tracings based on noise. Ideally an objective definition would be applied.

Response: Thank you for your diligence. The exclusion of data is unrelated to CHD or any clinical information. We have revised Figure 4, the "Data sources" paragraph of Methodology section, and the Supplementary materials to clarify the reasons. We have changed "case with corrupted diagnostic labels" to "case with missing diagnostic label" because the ECG signals and diagnostic labels are stored in two separate databases, and we have found that some of the

mappings are missing, resulting in unavailable diagnostic labels. We have also defined the excluded "corrupted ECG data" in the revised manuscript, which refers to ECG data with over 20% signal amplitudes exceeding 5 mV (usually 1-4 mV) or with over 20% recorded values of 0. The signal corruption is common since young children generally do not cooperate with the examinations. Therefore, the reason for data unavailability is objective, and we have made every effort to retain as much data as possible.

11. The authors argue that a potential value of the CHDdECG model is better detection of CHD in resource-poor settings, but the model was not evaluated in such settings (i.e., it was evaluated in two large tertiary care centers). I would also acknowledge geographic specificity of train/test as a potential limitation to generalizability.

Response: Thank you for reminding us. In the initial version, we acknowledged that “the data distribution of CHDs can vary in different areas and times, which may cause somewhat different effects of CHDdECG, especially in situations when the subtype proportions are markedly different from our test set and external test set.” In the revised version, we have highlighted the geographic and time specificity by adding a sentence:

“The geographic specificity and fixed period of training and validation limits the assessment of generalizability.”

12. Minor comments

- **Frequent grammatical errors requiring correction**
- **Odd phrases that require revision, eg “highly trusty”, “not excessively selected”, “hand crafted”**
- **Can the authors confirm that the normalization parameters used for inference in both test sets represent those obtained from the training set?**
- **Was early stopping employed in training?**

Response: Thank you for reminding us. We carefully rechecked and rewrote parts of paper to make it grammatically correct and concise. We use the normalization parameters obtained from the training set. We did not use early stop, but trained and fine-tuned with fixed 20 epochs. We have made it clear in the revised manuscript, thank you!

To Reviewer #2:

1. The authors mentioned that they constructed two test sets, an independent test set and an external test set. Please clarify the data source of the first independent test set and the test set used in the comparison of sensitivity on detecting CHD-related malformations between CHDdECG and ECG cardiologists.

Response: The comparison is conducted on data sampled from the first independent test set from Ref-A. We have clarified it in the revised manuscript. Thank you for your diligence!

2. One thing worth noting is that the sensitivity scores of CHDdECG are generally higher than that of cardiologists among all the groups. And the score gaps are around 20%. This phenomenon may be caused by the biased training when the test set is collected from the same center as training set. Please add experiments to demonstrate the cases that the cardiologists accept the explainable evidence and change their decisions, especially on the samples misidentified by cardiologists but correctly identified by CHDdECG in the test set. These experiments will make the work more valuable and convincing.

Response: Thank you for your suggestion, we believe your suggested additional experiment will make the work more valuable and convincing.

Firstly, we followed the suggestion of Reviewer #1 and use NRI metric to compare the performances of CHDdECG and cardiologist, so we reorganized the comparison experiments. Then, we followed your suggestion and conduct the additional experiment. The performances are illustrated as below (we have added it to Fig. 1g). We described the result in the main text:

“Further, we picked out those cases (with the highlighted key ECG segments; see Fig.3 for example) misidentified by cardiologists but correctly identified by CHDdECG for cardiologists’ rechecking. Based on the prompt of CHDdECG, the dark blue bars in Fig.1g indicated that cardiologists change parts of their original judgements for the picked cases and thus improved their detection performances. However, the NRI of cardiologists’ rechecking remained inferior to CHDdECG, suggesting that some cases are still indistinguishable for cardiologists.”

3. In Figure 3, the authors showed explainable evidence for the four sub-types of CHD, and the authors selected specific heartbeats variant in morphology. To have an objective view on the explainable evidence, the authors should provide and discuss the gradient heatmap of the misidentified samples.

Response: We provided two misidentified samples in the revised manuscript (also see the two figures below). As also discussed in the revised manuscript, the left figure illustrated a segment of ECG signal that the trained CHDdECG failed to correctly identify as the waveform of ventricular septal defect. Although it appears similar to the Katz-Wachtel phenomenon which represents manifestations of ventricular septal defect, we found that its amplitude is much smaller (the maximum values are around $1000 \mu\text{V}$), representing an atypical form that the model did not prioritize. For waveforms in the right figure, CHDdECG might misidentify the double-humped form as the “notch” (w.r.t. the atrial septal defect), however, its ground truth diagnostic label is “non-CHD”. It is witnessed from (e) and (f) that, the waveforms associated with CHD (or a certain subtype) are diverse and therefore challenging to be detect. In general, CHDdECG can identify some congenital heart disease related waveforms and the visualization results are partially understandable to humans, despite some misjudgments of confusing waveforms (but human can also learned from such misjudgments).

4. For data pre-processing, the authors normalized the three types of features using z-score. However, this approach will eliminate true amplitude for QRS-complex, which is the

supportive evidence for some morphology anomaly. The authors should provide the reason for ignoring the amplitude information.

Response: Thank you. We normalized the data for better model training. Since the same features of different data are normalized with the same parameters, thus the absolute values were only linearly scaled and did not affect the pattern mining practice of the deep learning model. Please note that the deep learning models have no prior knowledge like rule-based approaches, thus the absolute values of amplitude information are relatively helpless.

5. There are several confused issues for the hand-crafted features.

The first is about the “findpeaks” method from Matlab. This function will annotate all the peaks and can be easily disturbed by noise. The authors then used the method with slope in [61] to locate the onsets and offsets of the QRS-complex, P and T waves. In a technical perspective, the location should be erratic using this method, especially on P waves. Please validate the wave localization method on the open-access database with annotations for QRS-complex, P and T waves. The following database can be helpful.

LUDB database: <https://physionet.org/content/ludb/1.0.1/>

QT database: <https://physionet.org/content/qtdb/1.0.0/>

The second issue is that the authors should discuss why X_c is much less important than X_e while the inference stage of CHDdECG, since the ECG segmentation method is defective. More specifically, to verify if the insensitivity of CHDdECG on X_c is caused by inaccurate localization of ECG waves.

The third issue is that it is inappropriate to deem X_c as the features representing expert knowledge, since X_c is a common feature extraction based on ECG segmentation. It does not involve the discrimination of clinical information using expert knowledge. A more accurate description for X_c should be the features based on ECG segmentation. The authors can prove the capacity in knowledge discovery of CHDdECG with X_c and the explainable evidence. However, the method in this paper should not be defined as deep learning integrated with “Human Knowledge”.

Response: Thank you for your careful comments. We would like to answer the three questions by three points as follows.

(1). We demonstrate the effectiveness of our proposed method in the wave localization from both visualization and metrics perspectives. As shown in Figure 2 and Figure 3 below, it is intuitive that the predicted localizations and ground truth localizations are highly consistent, on both the LUDB and QT databases that you recommended. We annotate location of waves on 100 samples used in this work. See the example case in Figure 4, the predicted localizations are also in line with the manual annotations.

See Table 6 below, the location prediction performances are measured by precision, recall, F1-score (following “Viktor Moskalenko, et al. Deep Learning for ECG Segmentation.”), it is obvious that predicted locations by our approach is quite trusty.

Figure 2. An example of wave location prediction result, on a sample from LUDB database. The top panel represents the predicted locations by our approach, while the bottom panel represents the ground truth locations. Marks in different colors and shapes indicate different kinds of waveforms.

Figure 3. An example of wave location prediction result, on a sample from QT database. The top panel represents the predicted locations by our approach, while the bottom panel represents the ground truth locations. Marks in different colors and shapes indicate different kinds of waveforms.

Figure 4. An example of wave location prediction result, on a sample used in this work. The top panel represents the predicted locations by our approach, while the bottom panel represents the ground truth locations. Marks in different colors and shapes indicate different kinds of waveforms.

Table 6. Location prediction results on three datasets (LUDB database, QT database and a subset of the dataset for this work, containing 100 samples).

Datasets	Metrics	P onset	P peak	P offset	QRS onset	R peak	QRS offset	T onset	T peak	T offset
LUDB database	Recall	0.9765	0.9891	0.9714	0.9957	1.0000	0.9961	0.9865	0.9943	0.9894
	Precision	0.9642	0.9801	0.9621	0.9812	0.9993	0.9891	0.9812	0.9802	0.9803
	F1-score	0.9703	0.9846	0.9667	0.9884	0.9996	0.9926	0.9838	0.9872	0.9848
QT database	Recall	0.9702	0.9921	0.9762	0.9967	1.0000	0.9987	0.9817	0.9926	0.9812
	Precision	0.9699	0.9897	0.9721	0.9831	1.0000	0.9901	0.9775	0.9812	0.9800
	F1-score	0.9700	0.9909	0.9741	0.9899	1.0000	0.9944	0.9796	0.9869	0.9806
A subset of the dataset for this work	Recall	0.9876	0.9991	0.9881	0.9993	1.0000	1.0000	0.9910	0.9968	0.9963
	Precision	0.9766	0.9899	0.9665	0.9932	1.0000	0.9993	0.9902	0.9888	0.9910
	F1-score	0.9821	0.9945	0.9772	0.9962	1.0000	0.9996	0.9906	0.9928	0.9936

(2). We showed that our waveform localization approach is highly reliable. Regarding the reason why X_e is much more important than X_c , we believe that this may be due to the inadequacy of conventional human-crafted features to provide sufficient information for identifying CHD, which could also partially explain why cardiologists perform worse compared to the CHDdECG model. Since the waveforms of CHD are subtle yet important, automatic pattern mining from

signals (i.e., X_e) is beneficial. Furthermore, the additional ablation implementation (in response to 6) also supports this point.

(3). Thank you for your thoughtful comment. We acknowledge that the term “human knowledge” may not be appropriate, and we would like to gently clarify that the “clinical features” are not solely based on ECG segmentation, but rather they are computed by rules that are widely used by humans in analyzing ECG signals. These rules are given based on clinically useful concepts that are accepted by doctors, as shown in Table 1 of the Supplementary Materials. Therefore, we would like to update the term “human knowledge” to “human concepts”, which better reflects the features computed by rules representing clinically useful concepts. We hope our revision can clarify your concerns.

6. Please add ablation study for utilizing X_e , X_w and X_c as feature input separately. And the authors should discuss the effectiveness and rationality of using the three feature extraction methods according to the ablation results.

Response: Thank you for your suggestion. We have added ablation study in the Supplementary Materials. The results are illustrated in Figure 5 below. The proposed CHDdECG utilizes three feature types for comprehensive prediction. We conducted ablation experiments on the test set to examine the contributions of various feature types in deploying CHDdECG, using all or parts of the feature types. The results are illustrated in Figure 5 below, which demonstrates that the model using all three feature types performs best in general. Specifically, the recall rate is critical in CHD detection, and we observed that CHDdECG using all three feature types achieves considerably better recall rates than other settings. Additionally, it is noteworthy that the model performances with only wave features, clinical features, or ECG signals are all not bad, suggesting that all three feature types are informative. It is also observed that the model using only ECG signals outperforms the model using only clinical features, which further outperforms the model using only wavelet features. This performance ranking is consistent with the feature importance in the main paper.

Figure 5. An illustration of ablation study results, measured by recall (), specificity (), ROC-AUC (), brier score (). The legend indicates the used feature types in comparison, where “sig”, “clin”, and “wave” denote “ECG signals”, “clinical features”, and “wavelet features”, respectively.

Thank you for your helpful comments again. We believe that the revisions improved the article.

Your kind and early reply will be appreciated .

Thank you very much for your attention and consideration.

Kind regards,

Huiying Liang

On behalf of the other authors

2023-03-16

Reviewers' Comments:

Reviewer #1:

Remarks to the Author:

Most of my concerns have been addressed, and I thank the authors for their changes, which have greatly improved the clarity and potential impact of the work.

I have a few remaining comments:

1. Abstract – It is stated that the automatically extracted features from the electrocardiogram “were about 8 times more significant” than human concept features. This statement is incredibly vague. I would instead explain that the feature scores suggested 8 times greater influence of automatic ECG features on CHD predictions.
2. I appreciate the author’s inclusion of an NRI metric (at my suggestion) for the comparison to the clinical cardiologists. The NRI is a useful summary measure, but it is overly simplifying to present it in isolation. Rather, it is recommended to present both the NRI+ and the NRI-, which are the two components of the NRI calculation. This allows the reader to interpret whether the model gains are related to increased sensitivity, increased specificity, or both. Therefore, I suggest that parallel graphs showing NRI+ and NRI- are displayed, either in addition to the NRI as supplemental figures, or as a larger combined figure.
3. Table 1 – The title of the last column (the Probabilistic Predictions) needs to be updated for the lower panel.
4. Although I agree the disease labels used in this study are likely of good quality, I would nevertheless acknowledge the possibility of label misclassification as a limitation. As the authors argue, CHD can be present with ECG abnormalities below the level of human detection, and since many patients were included after having ECGs performed for non-cardiac indications (e.g., surgery), it is possible that some of these patients labeled as non-CHD actually had CHD that was never diagnosed because their ECGs had subtle changes (or no changes), which failed to trigger additional real-world evaluation. I acknowledge this limitation cannot be addressed without a prospective protocol dictating a standard diagnostic workup for all individuals, which would certainly be beyond the scope of this work, but it is nonetheless worth citing.

Reviewer #3:

Remarks to the Author:

I have had the privilege of reviewing this impressive article. The authors present an artificial intelligence (AI) model designed to detect congenital heart disease using electrocardiography. From my observation, the study seems to have been meticulously conducted, and the manuscript is comprehensively written with clear presentation. However, I would like to draw the authors' attention to a few points that need further attention and clarification:

1. The developed model's accuracy should be checked further in terms of stratification by patient age, sex, and heart rate. Additionally, the authors should provide a baseline characteristics table for the patient cohort.
2. A more precise external validation is necessary. I suggest obtaining and using external validation data from another country or a different device company. Alongside AUROC, the authors should also describe the area under the precision-recall curve (AUPRC).
3. A comprehensive table outlining the baseline characteristics of the patients is required. This table should encompass data on patients' epidemiology, ECG features, and echocardiographic features.
4. I would like the authors to verify whether any patients from the external validation data set were included in the training data set. If so, how can this be ensured?
5. The authors are requested to elucidate the process of model structure selection and the training process for the final model. Can the authors affirm with certainty that, among various models or structures considered, the final model selected is indeed the most accurate? Furthermore, should

the deep learning model using waves prove to be more advanced, the necessity of employing wavelets of clinical features may be eliminated. In this context, the authors need to ponder whether such a process could lead to overfitting accuracy.

Your further clarification on these issues would significantly enhance the manuscript. I look forward to your responses.

RESPONSE TO REVIEWERS

Of “Superior Detection of Congenital Heart Diseases by Pediatric
Electrocardiogram Based Deep Learning Integrated with Human Concepts”

Dear reviewers,

We wish to express our deep appreciation for granting us the opportunity to refine our manuscript. Your insights and suggestions are truly constructive, greatly enhancing the quality of our paper. We have carefully considered your feedback and diligently made the necessary revisions. The detailed responses to reviewers’ comments are provided in this document point by point. If there are any further concerns about our revised manuscript, please do not hesitate to let us know and we will promptly address them.

Huiying Liang

On behalf of the other authors

2023-08-31

To Reviewer #1

Comments 1: Abstract – It is stated that the automatically extracted features from the electrocardiogram “were about 8 times more significant” than human concept features. This statement is incredibly vague. I would instead explain that the feature scores suggested 8 times greater influence of automatic ECG features on CHD predictions.

Response: Thank you for your thoughtful suggestion. We fully acknowledge the vagueness of the previous statement. Following your recommendations, we have revised the sentence from: “*the automatically extracted features from electrocardiogram were about 8 times more significant than human concept features*” to “*feature importance scores suggested 8 times greater influence of automatically extracted electrocardiogram features on CHD detection compared with human concept features*”.

Comments 2: I appreciate the author’s inclusion of an NRI metric (at my suggestion) for the comparison to the clinical cardiologists. The NRI is a useful summary measure, but it is overly simplifying to present it in isolation. Rather, it is recommended to present both the NRI+ and the NRI-, which are the two components of the NRI calculation. This allows the reader to interpret whether the model gains are related to increased sensitivity, increased specificity, or both. Therefore, I suggest that parallel graphs showing NRI+ and NRI- are displayed, either in addition to the NRI as supplemental figures, or as a larger combined figure.

Response: Thank you very much for further providing detailed suggestions on NRI metric. We wholeheartedly agree that analyzing NRI+ and NRI- can provide additional insight to interpret whether the model gains are related to increased sensitivity, increased specificity, or both. We included the results of both NRI(+) and NRI(-) in Figure S2 of the Supplementary Materials (see below). We analyze the results incorporating NRI, NRI(+), and NRI(-) in Supplementary Materials section, reading as follows:

“As shown in Figure S2(a), the NRI scores of the CHDdECG model and cardiologists assisted by CHDdECG identified activated segments, compared with the performance

of cardiologists without any assistance as the benchmark, indicate that CHDdECG demonstrates superior CHD detection capabilities in comparison to cardiologists and demonstrated that CHDdECG has the potential to enhance the diagnostic performance of cardiologists. The NRI(+) and NRI(-) scores are depicted in Figure S2 (b) and (c), respectively. NRI(+) scores reveal a significant contribution in enhancing the NRI values. Combined with the marginal NRI(-) scores, NRI analysis suggests that the high performance of CHDdECG is predominantly attributed to its proficiency in identifying CHD cases (increased sensitivity).”

Figure S2. (a) illustrated NRI of CHDdECG and cardiologists assisted by CHDdECG, compared with cardiologists without any assistance as benchmark, across 10 randomly sampled test data groups from the Center-A test set. (b) and (c) illustrated the corresponding NRI(+) and NRI(-).

Comments 3: Table 1 – The title of the last column (the Probabilistic Predictions) needs to be updated for the lower panel.

Response: Thank you for your careful suggestions. We consulted a statistics expert for thorough revisions, and have updated the last column of Table 1. The revised Table 1

is presented in the manuscript as below.

Test Set from Center-A						
Categories	Prop(%)	ROC-AUC	Spec	Sens	Brier	Probabilistic Predictions
Atrial septal defect	12.2%	0.835	0.910	0.504	0.0736	
Patent ductus arteriosus	6.7%	0.856	0.912	0.522	0.0673	
Anomalous origin of a coronary artery	0.9%	0.894	0.938	0.722	0.0459	
Ventricular septal defect	36.6%	0.920	0.925	0.707	0.0648	
Coarctation of the aorta	1.9%	0.935	0.935	0.763	0.0487	
Total anomalous pulmonary venous connection	0.8%	0.944	0.946	0.706	0.0385	
dextro-Transposition of the great arteries	2.1%	0.929	0.952	0.721	0.0349	
Pulmonary atresia	2.5%	0.985	0.966	0.902	0.0255	
Single ventricle	0.9%	0.985	0.959	0.947	0.0284	
Tetralogy of fallot	3.3%	0.987	0.966	0.927	0.0256	
Atrioventricular septal defect	2.0%	0.991	0.946	0.974	0.0372	
Double-outlet right ventricle	0.8%	0.992	0.963	1.000	0.0267	
CHD (with all sub-types)	-	0.915	0.881	0.800	0.0998	

External Test Set from Center-B						
Categories	Prop(%)	ROC-AUC	Spec	Sens	Brier	Probabilistic Predictions
Atrial septal defect	28.3%	0.926	0.958	0.694	0.0357	
Patent ductus arteriosus	12.0%	0.889	0.961	0.500	0.0325	
Ventricular septal defect	33.3%	0.918	0.969	0.670	0.0296	
CHD (with all sub-types)	-	0.917	0.937	0.770	0.0568	

External Test Set from Center-C						
Categories	Prop(%)	ROC-AUC	Spec	Sens	Brier	Probabilistic Predictions
Atrial septal defect	52.2%	0.916	0.936	0.705	0.0825	
Patent ductus arteriosus	7.9%	0.904	0.938	0.673	0.0538	
Anomalous origin of a coronary artery	0.6%	0.876	0.960	0.533	0.0340	
Ventricular septal defect	23.4%	0.913	0.948	0.632	0.0616	
Coarctation of the aorta	1.0%	0.859	0.956	0.375	0.0365	
Pulmonary atresia	2.3%	0.906	0.976	0.842	0.0221	
Single ventricle	0.6%	0.929	0.973	0.733	0.0203	
Tetralogy of fallot	3.7%	0.939	0.976	0.837	0.0216	
Atrioventricular septal defect	0.7%	0.909	0.966	0.611	0.0279	
CHD (with all sub-types)	-	0.907	0.907	0.786	0.1043	

Table 1. Model performances on the internal test set from Center-A, an external test set from Center-B, and another external test set from Center-C. The reported performances for detecting CHDs or non-CHD (marked in blue) were obtained with the CHDdECG model trained from scratch, while the performances for various CHD subtypes were obtained separately after the CHDdECG model was specifically fine-tuned. Notably, some subtypes were not included since their amounts were less than 0.5% of all the CHD cases or less than 10. “Prop”, “Spec”, “Sens” and “Brier” indicate “Proportion”, “Specificity”, “Sensitivity” and “Brier Score”, respectively. The probabilistic predictions are represented by box-plots for better viewing the distributions. Box-plot elements are defined as: center line, median; box limits, upper and lower quartiles; whiskers, 1.5× interquartile range.

Comments 4: Although I agree the disease labels used in this study are likely of good quality, I would nevertheless acknowledge the possibility of label misclassification as a limitation. As the authors argue, CHD can be present with ECG abnormalities below the level of human detection, and since many patients were included after having ECGs performed for non-cardiac indications (e.g., surgery), it is possible that some of these patients labeled as non-CHD actually had CHD that was never diagnosed because their ECGs had subtle changes (or no changes), which failed to trigger additional real-world evaluation. I acknowledge this limitation cannot be addressed without a prospective protocol dictating a standard diagnostic workup for all

individuals, which would certainly be beyond the scope of this work, but it is nonetheless worth citing.

Response: Thank you for your perceptive and constructive suggestion. We wholeheartedly acknowledge the possibility of label misclassification even we have tried our best to follow standardized diagnostic guidelines and exclude the case after intervention. Thus, we included this point as a limitation in the discussion section, reading as follows:

“.....Fifth, although the CHD labels were acquired following standardized diagnostic guidelines, we cannot rule out the possibility of label misclassification as a limitation, particularly when CHD cases present with abnormalities below the level of human detection. This limitation also highlights the need of a prospective protocol research dictating a comprehensive and standard diagnostic workup for all individuals.”

To Reviewer #3

Comments 1: The developed model's accuracy should be checked further in terms of stratification by patient age, sex, and heart rate. Additionally, the authors should provide a baseline characteristics table for the patient cohort.

Response: We fully agree and appreciate your helpful suggestions. This allows the reader more directly understand the potential practice scope and boundaries of the model. Thus, we reported the model performances stratified by patient age, sex, and heart rate in Table S5 of the Supplementary Materials (presents as below). More comprehensive evaluation was also conducted on various metrics including ROC-AUC, PR-AUC (following your next comments), specificity, sensitivity, and brier score. Corresponding results were added in the Supplementary Materials section, reading as follows:

Test Set from Center-A								
Characteristic	Stratification	Count	Prop	ROC-AUC	PR-AUC	Spec	Sens	Brier
Sex	male	7758	64.65%	0.931	0.764	0.907	0.805	0.0837
	female	4242	35.35%	0.918	0.764	0.904	0.791	0.0887
Age (year)	< 1	2876	23.97%	0.936	0.773	0.894	0.824	0.0873
	1-2	2648	22.07%	0.922	0.785	0.917	0.797	0.0826
	2-3	1829	15.24%	0.907	0.719	0.911	0.755	0.0896
	3-4	4647	38.73%	0.931	0.768	0.904	0.805	0.0844
Heart rate (bpm)	< 60	233	1.94%	0.954	0.643	0.915	0.844	0.0731
	60-80	336	2.80%	0.892	0.737	0.855	0.803	0.1248
	81-100	1651	13.76%	0.898	0.699	0.853	0.781	0.1158
	101-120	2982	24.85%	0.921	0.764	0.906	0.793	0.0855
	121-140	3035	25.29%	0.935	0.783	0.916	0.803	0.0800
	141-160	2442	20.35%	0.935	0.798	0.924	0.824	0.0722
	> 160	1321	11.01%	0.934	0.758	0.920	0.781	0.0769

External Test Set from Center-B								
Characteristic	Stratification	Count	Prop	ROC-AUC	PR-AUC	Spec	Sens	Brier
Sex	male	3679	51.55%	0.929	0.487	0.933	0.803	0.0576
	female	3458	48.45%	0.902	0.446	0.940	0.732	0.0562
Age (year)	< 1	2171	35.46%	0.949	0.479	0.937	0.819	0.0542
	1-2	1871	24.93%	0.903	0.429	0.932	0.705	0.0607
	2-3	1998	21.27%	0.905	0.525	0.943	0.754	0.0545
	3-4	2081	18.34%	0.897	0.460	0.935	0.794	0.0596
Heart rate (bpm)	< 60	99	0.21%	0.923	0.417	0.923	1.000	0.0808
	60-80	338	1.61%	0.896	0.817	0.921	0.564	0.1294
	81-100	1502	7.97%	0.892	0.491	0.937	0.714	0.0649
	101-120	2022	17.46%	0.898	0.495	0.934	0.746	0.0619
	121-140	1813	27.03%	0.938	0.472	0.938	0.883	0.0551
	141-160	1463	27.84%	0.925	0.434	0.935	0.800	0.0528
	> 160	884	17.88%	0.925	0.480	0.941	0.844	0.0506

External Test Set from Center-C								
Characteristic	Stratification	Count	Prop	ROC-AUC	PR-AUC	Spec	Sens	Brier
Sex	male	4398	54.16%	0.903	0.802	0.906	0.776	0.1086
	female	3723	45.84%	0.913	0.832	0.908	0.798	0.0993
Age (year)	< 1	2171	26.73%	0.900	0.812	0.906	0.792	0.1066
	1-2	1871	23.04%	0.907	0.810	0.897	0.786	0.1058
	2-3	1998	24.60%	0.907	0.803	0.903	0.787	0.1051
	3-4	2081	25.62%	0.916	0.833	0.921	0.777	0.0999
Heart rate (bpm)	< 60	99	1.22%	0.843	0.547	0.851	0.800	0.1030
	60-80	338	4.16%	0.885	0.848	0.774	0.885	0.1330
	81-100	1502	18.50%	0.898	0.867	0.876	0.823	0.1204
	101-120	2022	24.90%	0.909	0.812	0.913	0.794	0.0951
	121-140	1813	22.32%	0.898	0.751	0.913	0.742	0.1037
	141-160	1463	18.02%	0.912	0.799	0.921	0.762	0.0995
	> 160	884	10.89%	0.917	0.848	0.944	0.730	0.0966

Table S5. Stratification analyses of individuals and CHDdECG's detection effect verification on an internal and two external test sets. ROC-AUC: receiver operating characteristic – area under the curve; PR-AUC: precision-recall – area under the curve; Spec: specificity; Sens: sensitivity; Brier: brier score.

“The CHDdECG’s performance was checked further in terms of stratification by patient age, sex, and heart rates (Table S5). On the internal and two external test sets, the CHDdECG model consistently excels across sex, age, and heart rate stratifications, demonstrating a uniform and robust level of effectiveness in diverse characteristics.”

We also agree that a comprehensive baseline characteristics table for the patient cohorts can clearly present essential demographic and clinical information about the patients included in a study. A baseline characteristics table is included in Table S2 of the Supplementary Materials, please refer to the responses to comments 3 below.

Comments 2: A more precise external validation is necessary. I suggest obtaining and using external validation data from another country or a different device company. Alongside AUROC, the authors should also describe the area under the precision-recall curve.

Response: Thank you and strongly agree with your suggestion to add verification for different regions and devices. Happily, besides the external test set in the Southern China (Guangzhou City, Guangdong Province), we have successfully acquired support from a large children's medical center in the Northeast China (Shengyang City, Liaoning Province, nearly 2300 kilometers away from Guangzhou) with different ECG devices (GE MAC800 in Guangzhou, NIHON KOHDEN ECG-2550 in Shenyang). Of course, we have thoroughly added the detailed information about the new external test set in corresponding data collection, result description, and discussion sections, see the revised manuscript.

According to your helpful suggestion, alongside AUROC, we also describe the area under the precision-recall curve across test set from Center A to Center-C in Figure 2, Table S4, and Table S5, respectively, as below.

Figure 2 (e) and (f), illustrating the ROC and PR curves of CHDdECG on three test sets.

Test Set from Center-A						
CHD Subtypes	CHDdECG	CNN	LSTM	k-NN	RF	Xgboost
Atrial septal defect	0.130	0.088	0.128	0.097	0.132	0.073
Patent ductus arteriosus	0.063	0.028	0.066	0.032	0.084	0.067
Anomalous origin of a coronary artery	0.027	0.020	0.025	0.027	0.020	0.011
Ventricular septal defect	0.540	0.362	0.346	0.141	0.291	0.395
Coarctation of the aorta	0.080	0.053	0.062	0.041	0.074	0.047
Total anomalous pulmonary venous connection	0.054	0.035	0.063	0.056	0.050	0.046
dextro-Transposition of the great arteries	0.096	0.038	0.071	0.065	0.043	0.023
Pulmonary atresia	0.233	0.136	0.148	0.113	0.157	0.133
Single ventricle	0.082	0.052	0.047	0.050	0.058	0.083
Tetralogy of fallot	0.361	0.223	0.241	0.014	0.320	0.242
Atrioventricular septal defect	0.241	0.152	0.175	0.096	0.173	0.049
Double-outlet right ventricle	0.160	0.166	0.085	0.071	0.066	0.102
CHD (with all the sub-types)	0.721	0.663	0.692	0.270	0.652	0.586

External Test Set from Center-B						
CHD Subtypes	CHDdECG	CNN	LSTM	k-NN	RF	Xgboost
Atrial septal defect	0.264	0.278	0.218	0.091	0.221	0.221
Patent ductus arteriosus	0.114	0.092	0.094	0.041	0.144	0.152
Ventricular septal defect	0.288	0.256	0.197	0.079	0.202	0.266
CHD (with all the sub-types)	0.464	0.424	0.413	0.132	0.412	0.461

External Test Set from Center-C						
CHD Subtypes	CHDdECG	CNN	LSTM	k-NN	RF	Xgboost
Atrial septal defect	0.717	0.665	0.686	0.501	0.696	0.585
Patent ductus arteriosus	0.303	0.265	0.291	0.128	0.252	0.208
Anomalous origin of a coronary artery	0.043	0.056	0.096	0.105	0.065	0.043
Ventricular septal defect	0.531	0.492	0.398	0.286	0.431	0.436
Coarctation of the aorta	0.035	0.085	0.071	0.028	0.021	0.014
Pulmonary atresia	0.119	0.103	0.148	0.103	0.096	0.103
Single ventricle	0.097	0.090	0.062	0.065	0.102	0.082
Tetralogy of fallot	0.388	0.312	0.343	0.193	0.298	0.395
Atrioventricular septal defect	0.060	0.057	0.035	0.047	0.028	0.041
CHD (with all the sub-types)	0.815	0.698	0.732	0.480	0.682	0.610

Table S4. Classification performances measured by "PR-AUC" on internal and external test sets.

Test Set from Center-A								
Characteristic	Stratification	Count	Prop	ROC-AUC	PR-AUC	Spec	Sens	Brier
Sex	male	7758	64.65%	0.931	0.764	0.907	0.805	0.0837
	female	4242	35.35%	0.918	0.764	0.904	0.791	0.0887
Age (year)	< 1	2876	23.97%	0.936	0.773	0.894	0.824	0.0873
	1-2	2648	22.07%	0.922	0.785	0.917	0.797	0.0826
	2-3	1829	15.24%	0.907	0.719	0.911	0.755	0.0896
	3-4	4647	38.73%	0.931	0.768	0.904	0.805	0.0844
Heart rate (bpm)	< 60	233	1.94%	0.954	0.643	0.915	0.844	0.0731
	60-80	336	2.80%	0.892	0.737	0.855	0.803	0.1248
	81-100	1651	13.76%	0.898	0.699	0.853	0.781	0.1158
	101-120	2982	24.85%	0.921	0.764	0.906	0.793	0.0855
	121-140	3035	25.29%	0.935	0.783	0.916	0.803	0.0800
	141-160	2442	20.35%	0.935	0.798	0.924	0.824	0.0722
	> 160	1321	11.01%	0.934	0.758	0.920	0.781	0.0769

External Test Set from Center-B								
Characteristic	Stratification	Count	Prop	ROC-AUC	PR-AUC	Spec	Sens	Brier
Sex	male	3679	51.55%	0.929	0.487	0.933	0.803	0.0576
	female	3458	48.45%	0.902	0.446	0.940	0.732	0.0562
Age (year)	< 1	2171	35.46%	0.949	0.479	0.937	0.819	0.0542
	1-2	1871	24.93%	0.903	0.429	0.932	0.705	0.0607
	2-3	1998	21.27%	0.905	0.525	0.943	0.754	0.0545
	3-4	2081	18.34%	0.897	0.460	0.935	0.794	0.0596
Heart rate (bpm)	< 60	99	0.21%	0.923	0.417	0.923	1.000	0.0808
	60-80	338	1.61%	0.896	0.817	0.921	0.564	0.1294
	81-100	1502	7.97%	0.892	0.491	0.937	0.714	0.0649
	101-120	2022	17.46%	0.898	0.495	0.934	0.746	0.0619
	121-140	1813	27.03%	0.938	0.472	0.938	0.883	0.0551
	141-160	1463	27.84%	0.925	0.434	0.935	0.800	0.0528
	> 160	884	17.88%	0.925	0.480	0.941	0.844	0.0506

External Test Set from Center-C								
Characteristic	Stratification	Count	Prop	ROC-AUC	PR-AUC	Spec	Sens	Brier
Sex	male	4398	54.16%	0.903	0.802	0.906	0.776	0.1086
	female	3723	45.84%	0.913	0.832	0.908	0.798	0.0993
Age (year)	< 1	2171	26.73%	0.900	0.812	0.906	0.792	0.1066
	1-2	1871	23.04%	0.907	0.810	0.897	0.786	0.1058
	2-3	1998	24.60%	0.907	0.803	0.903	0.787	0.1051
	3-4	2081	25.62%	0.916	0.833	0.921	0.777	0.0999
Heart rate (bpm)	< 60	99	1.22%	0.843	0.547	0.851	0.800	0.1030
	60-80	338	4.16%	0.885	0.848	0.774	0.885	0.1330
	81-100	1502	18.50%	0.898	0.867	0.876	0.823	0.1204
	101-120	2022	24.90%	0.909	0.812	0.913	0.794	0.0951
	121-140	1813	22.32%	0.898	0.751	0.913	0.742	0.1037
	141-160	1463	18.02%	0.912	0.799	0.921	0.762	0.0995
	> 160	884	10.89%	0.917	0.848	0.944	0.730	0.0966

Table S5. Stratification analyses of individuals and CHDdECG's detection effect verification on an internal and two external test sets. ROC-AUC: receiver operating characteristic – area under the curve; PR-AUC: precision-recall – area under the curve; Spec: specificity; Sens: sensitivity; Brier: brier score.

What's even more interesting, we have noted that PR-AUC performances display

greater variability in comparison to other metrics. After consultations with statistical experts, we have identified that this phenomenon stems from the label imbalance inherent to the problem. We discuss this phenomenon in the discussion section, reading as follows:

“While CHDdECG’s performances on Center-B appear notable in terms of ROC-AUC, specificity, and sensitivity, Fig.1 (f) uncovers a relatively low PR-AUC value of around 0.5. Conversely, the PR-AUC on Center-C (though not perfectly balanced) showcases robust performance, surpassing 0.8. This phenomenon can be attributed to the significant label imbalance present in Center-B.”

Comments 3: A comprehensive table outlining the baseline characteristics of the patients is required. This table should encompass data on patients' epidemiology, ECG features, and echocardiographic features.

Response: Thank you for your suggestions to make the article more readable. Together with your helpful comments 1, We provided a comprehensive table outlining the baseline characteristics of the patients in terms of age, sex, ECG features and echocardiographic features, as shown in Table S2 of Supplementary Materials. Additionally, we provide a succinct overview of characteristic distributions with the statement as follows:

Characteristics		Training & validation set from Center-A	Test set from Center-A	External test set from Center-B	External test set from Center-C
Age (years):		2.12 ± 1.50	2.13 ± 1.23	1.55 ± 1.04	1.95 ± 1.17
Sex:	Male	4196 (63.8%)	7758 (64.65%)	3679 (51.55%)	4398 (54.16%)
	Female	23873 (36.2%)	4242 (35.35%)	3458 (48.45%)	3723 (45.84%)
ECG Features:	P wave apex values of lead I (uV)	47.2±30.6	47.4±30.8	40.4±24.8	63.6±42.2
	P wave apex values of lead II (uV)	47.2±28.5	47.2±29.2	42.1±24.1	66±46.3
	P wave apex values of lead III (uV)	32.7±24.9	32.3±25	28.4±20.3	43.9±36.2
	P wave apex values of lead aVR (uV)	62.9±25.3	63±25.7	59±21.6	84.7±45
	P wave apex values of lead aVL (uV)	35±27.3	34.8±27.5	29.8±21.4	46.4±35.4
	P wave apex values of lead aVF (uV)	35.8±25.3	35.2±24.9	31.9±21.4	48.4±37.8
	P wave apex values of lead V1 (uV)	69.7±30.8	70.2±30.9	62.9±26.1	85±42.5
	P wave apex values of lead V3 (uV)	53.7±48.8	53.4±48.7	23.9±45.4	63.1±60.4
	P wave apex values of lead V5 (uV)	19.7±55.4	19.8±55.5	5.7±42.6	23.1±83.7
	Q wave apex values of lead I (uV)	-67.8±35.7	-68±36	-58.7±27.1	-85.3±51.7
	Q wave apex values of lead II (uV)	-111.7±48.8	-112.4±49.4	-111.1±45.2	-138.3±70.9
	Q wave apex values of lead III (uV)	-117±74.8	-118.5±75.4	-112.3±68.4	-153.4±116.3
	Q wave apex values of lead aVR (uV)	-223.1±116.7	-221.2±115.9	-204.2±108.9	-433.3±352.5
	Q wave apex values of lead aVL (uV)	-46.1±37.6	-45±36.2	-35.3±24.9	-57.6±52.4
	Q wave apex values of lead aVF (uV)	-102.4±56.7	-103.2±57.6	-102.4±53.8	-125.9±77.4
	Q wave apex values of lead V1 (uV)	-48.8±39.2	-48.2±39.5	-39.7±30.1	-63.8±55.4
	Q wave apex values of lead V3 (uV)	-100.8±56.5	-101.3±57.2	-111.5±50.8	-123.2±79.4
	Q wave apex values of lead V5 (uV)	-148.9±71	-148.7±70.3	-128±53.4	-180.7±97.1
	R wave apex values of lead I (uV)	267.4±109.4	269.4±111.4	222.1±90.8	448.7±309.9
	R wave apex values of lead II (uV)	399.3±148.3	400±149.7	376.2±142.6	628.4±388.1
	R wave apex values of lead III (uV)	323.3±160.7	325.5±161.6	301±149.2	454.2±289.1
	R wave apex values of lead aVR (uV)	180.5±75.5	181±76.9	150.6±53.6	294.9±194.6
	R wave apex values of lead aVL (uV)	174.7±90.4	175.9±90.1	135.4±62.5	272.6±195.5
	R wave apex values of lead aVF (uV)	335.2±148	336.1±149	318.1±140.8	487.5±298
	R wave apex values of lead V1 (uV)	435.9±180.3	438.1±181.4	430±166.6	610.4±374.4
	R wave apex values of lead V3 (uV)	770.2±230.5	771.1±228.4	789±222.5	1261.3±803.1
	R wave apex values of lead V5 (uV)	653.5±229.9	650.1±226.2	533±186	1048.4±672.5
	S wave apex values of lead I (uV)	-171.4±90.9	-171.2±90.6	-154.7±76.8	-285.6±225.4
	S wave apex values of lead II (uV)	-150.7±62.8	-149.9±62.4	-138.8±54.6	-265.6±195.9
	S wave apex values of lead III (uV)	-77.7±39.8	-77.5±40.6	-69±31.4	-128.8±96.2
	S wave apex values of lead aVR (uV)	-72.8±91.7	-74.6±92	-63.8±81.8	-65.2±85.3
	S wave apex values of lead aVL (uV)	-166.1±111	-167.6±112	-159.6±102.4	-241.8±200.4
	S wave apex values of lead aVF (uV)	-102.5±45.2	-102±45	-93.7±38.1	-174.3±128
	S wave apex values of lead V1 (uV)	-337.8±187.7	-340.1±189.2	-325.5±169.3	-443.7±289.5
	S wave apex values of lead V3 (uV)	-573.7±255	-571.1±252.2	-505.5±214.5	-931.6±656
	S wave apex values of lead V5 (uV)	-281.3±126.2	-280.4±125.2	-224.5±89.2	-489.9±374
	T wave apex values of lead I (uV)	78.3±42.4	77.9±42.5	66.8±38.2	107.2±62
	T wave apex values of lead II (uV)	90±43.4	90.2±43.8	81.7±40	120.1±67.2
	T wave apex values of lead III (uV)	29±29.5	29.1±30.2	20.7±24.1	46.7±48.3
	T wave apex values of lead aVR (uV)	45±28.5	45±28.6	38.7±24.8	60.3±36
	T wave apex values of lead aVL (uV)	44.5±27.4	44.2±27.1	35.1±19.1	67.9±48.5
	T wave apex values of lead aVF (uV)	53±33.7	53±33.9	46.4±29.5	75±52.4
	T wave apex values of lead V1 (uV)	55±43.9	54.7±44.8	41±42.2	78±66
	T wave apex values of lead V3 (uV)	84.8±54.8	84.1±55.4	81.6±53.4	125.6±93.3
	T wave apex values of lead V5 (uV)	124±74.2	124.7±73.9	107.6±59.1	163.5±96.7
	P-P interval duration (ms)	488±99.8	487.2±100.2	447.6±82.6	505.6±113.8
	QRS complex duration (ms)	168±40.2	169±40.8	163.8±38	148.4±45
R-R interval duration (ms)	488.4±100.2	486.8±99.8	447.1±81.8	505.6±113.8	
S-T interval duration (ms)	127±43	126.4±43.2	135.6±40.4	151.2±55.6	
ST segment duration (ms)	44.8±31.4	44.2±31.4	20.8±15.8	46.2±28.6	
Q-T interval duration (ms)	297.8±36.2	298.2±37.4	303±32.8	302.4±38.6	
P-R interval duration (ms)	77.6±29.2	78.4±29.6	89.2±29.4	88.6±35.6	
PR segment duration (ms)	25±16.6	25±16.6	15.4±13	25.1±16.6	
T-T interval duration (ms)	488±99.4	487±99.4	446.8±81.6	505.2±113.4	
duration of P wave (ms)	58.4±27.8	58.8±27.8	87.8±28.6	67.6±29.8	
duration of T wave (ms)	87.6±18.6	87.6±18.6	128.6±25.2	107.8±36.6	
Echocardiographic Features:	Atrial septal defect	1311 (12%)	248 (12.2%)	85 (28.3%)	1315 (52.2%)
	Patent ductus arteriosus	710 (6.5%)	136 (6.7%)	36 (12%)	199 (7.9%)
	Anomalous origin of a coronary artery	68 (0.6%)	18 (0.9%)	/	15 (0.6%)
	Ventricular septal defect	3975 (36.4%)	745 (36.6%)	100 (33.3%)	589 (23.4%)
	Coarctation of the aorta	246 (2.3%)	38 (1.9%)	/	24 (1%)
	Total anomalous pulmonary venous connection	141 (1.3%)	17 (0.8%)	/	9 (0.4%)
	dextro-Transposition of the great arteries	270 (2.5%)	43 (2.1%)	2 (0.7%)	11 (0.4%)
	Pulmonary atresia	248 (2.3%)	51 (2.5%)	3 (1.0%)	37 (2.3%)
	Single ventricle	106 (1%)	19 (0.9%)	/	15 (0.6%)
	Tetralogy of fallot	370 (3.4%)	68 (3.3%)	/	92 (3.6%)
	Atrioventricular septal defect	169 (1.5%)	40 (2.0%)	/	18 (0.7%)
	Double-outlet right ventricle	101 (0.9%)	16 (0.8%)	2 (0.7%)	9 (0.4%)
	Congenital Mitral Valve Insufficiency	34 (0.3%)	10 (0.5%)	2 (0.7%)	8 (0.3%)
	Congenital Aortic Arch Disruption	33 (0.3%)	11 (0.5%)	/	6 (0.2%)
	Tricuspid Atresia	26 (0.2%)	7 (0.3%)	/	/
	Common Arterial Trunk	25 (0.2%)	5 (0.2%)	4 (1.3%)	/
	Congenital Aortic Valve Stenosis	13 (0.1%)	3 (0.1%)	/	5 (0.2%)
	Ebstein's anomaly	11 (0.1%)	3 (0.1%)	/	5 (0.2%)
	Congenital Tricuspid Valve Insufficiency	12 (0.1%)	2 (0.1%)	/	/
	double chamber right ventricle	10 (0.1%)	2 (0.1%)	/	6 (0.2%)
	Coronary Artery to Right Ventricle Fistula	8 (0.1%)	2 (0.1%)	2 (0.7%)	/
	Double Aortic Arch	7 (0.1%)	2 (0.1%)	/	9 (0.4%)
	Congenital Coronary Artery Anomaly	7 (0.1%)	2 (0.1%)	/	/
	Congenital Tricuspid Valve Atresia	6 (0.1%)	2 (0.1%)	/	/
	Ventricular Noncompaction Cardiomyopathy	6 (0.1%)	2 (0.1%)	/	/
	Falot's Pentalogy	6 (0.1%)	2 (0.1%)	/	5 (0.2%)
	Coronary Artery to Right Atrium Fistula	6 (0.1%)	2 (0.1%)	/	8 (0.3%)
	Congenital Coronary Artery to Pulmonary Artery Fistula	6 (0.1%)	2 (0.1%)	2 (0.7%)	9 (0.4%)
	unknown sub-types	2996 (27.4%)	540 (26.5%)	62 (20.7%)	107 (2.2%)

Table S2. Demographic and clinical characteristics of cohorts. Values are represented in mean ± standard deviation (minimum–maximum) or count (proportion, %).

“After excluding cases based on the aforementioned criteria, our cohort consisted of 65,869 cases for model training and validation, the internal test set containing 12,000 cases, as well as the two external test sets containing 7,137 and 8,121 cases, respectively. We presented the demographic and clinical characteristics of these cohorts in Table S2, which provides a summary of distributions in term of age, sex, ECG features, and echocardiographic features. Basic demographic features such as age and sex showed good consistency across four different cohorts. Significant difference of echocardiographic diagnostic types from two externally independent test sets gives us the possibility to more broadly validate the CHDdECG model.”

Comments 4: I would like the authors to verify whether any patients from the external validation data set were included in the training data set. If so, how can this be ensured?

Response: We fully understand and agree with your concerns. Because ensuring that patients from the test data set were not included in the training data set is crucial for objective evaluations of the model.

Given the training data set of Center-A and the external test set of Center-B are from two different referral centers of the same tertiary hospital, we have excluded patients from the Center-B if they are already present in the Center-A data set by using the enterprise master patient index (EMPI) which includes factors such as age, date of birth, sex, and name. In data pre-processing, we have taken the necessary steps to exclude those patients from Center-B test set. For the external test set from Center-C, spatial distance and comparison of basic information are used to ensure the independence of sample composition. The clarification has been diligently integrated into the revised manuscript, see the Methodology section, reading as follows:

“.....In this study, a total of three data sets, including a comprehensive dataset used for model training, validation, and internal testing from Center-A, an external test set from Center-B, and another external test data set from Center-C, were used to develop and broadly test the model. Center-A and Center-B are two different referral centers from the same tertiary hospital located in Southern China. Center-C is a large children's medical center located in Northeast China. Furthermore, the ECG data in Center-A and Center-B were collected using identical ECG devices (GE MAC800) from August 2014 to October 2020, while the data in Center-C were collected utilizing a distinct brand of ECG device (NIHON KOHDEN ECG-2550) from January 2020 to June 2023.....”

“.....To ensure that patients from the test data sets were not included in the training data set, we excluded patients from the Center-B external test set if they are already present in the Center-A training set by using the enterprise master patient index (EMPI) which includes factors such as age, date of birth, sex, and name. For Center-C external test set, a distance of almost 2300 kilometers combined with comparison rules based on patients' name, sex, age, and CHD type was used to

ensure the absence of patient overlap with other centers.....”

Comments 5: The authors are requested to elucidate the process of model structure selection and the training process for the final model. Can the authors affirm with certainty that, among various models or structures considered, the final model selected is indeed the most accurate? Furthermore, should the deep learning model using waves prove to be more advanced, the necessity of employing wavelets of clinical features may be eliminated. In this context, the authors need to ponder whether such a process could lead to overfitting accuracy.

Response: We fully agree and thank you for your insightful comments. The process of model structure selection and the training process for the final model has been detailed within the Methodology section of the current manuscript.

(i) Regarding the model structure selection, our goal is to mine as many modality features as possible based on ECG examination to improve the accuracy of CHD detection. Raw ECG signal, clinical features presenting human concepts, and wavelet features are three key and representative kinds of modality features. To prove CHDdECG integrating three kinds of modality features is the most accurate, we conducted ablation study to inspect the contributions of each modality. Results are graphically represented in Figure. S1 of the Supplementary Materials (as shown below; in the legend, “clin”, “sig”, and “wave” indicates the “clinical features presenting human concepts”, “raw ECG signals”, and “wavelet features”, respectively).

Figure S1. An illustration of ablation study results, measured by recall (\uparrow), specificity (\uparrow), ROC-AUC (\uparrow), brier score (\downarrow). The legend indicates the used feature types in comparison, where “sig”, “clin”, and “wave” denote “ECG signals”, “clinical features”, and “wavelet features”, respectively.

The corresponding descriptions are demonstrated in Supplementary Materials, reading as follows:

“.....The model using all three feature types performs best in general. Specifically, the recall rate is critical in CHD detection, and we observed that CHDdECG using all three feature types achieves considerably better recall rates than other settings. Additionally, it is noteworthy that the model performances with only wave features, clinical features, or ECG signals are all not too bad, suggesting that all three feature types are informative.....”

(ii) Comparison with other classical models. As shown in Table S3 and Table S4 in the Supplementary Materials, we compared our designed CHDdECG with other classical models, including 1D convolutional neural network (CNN) following the setting of Hannun’s work ^[1], long-short term memory (LSTM) with default hyperparameters, k-nearest neighboring classifier, and XGboost model, and random forest model (RF). Results on three test sets measured with PR-AUC and ROC-AUC indicate that our CHDdECG outperforms those classical models, implying the superiority of the model structure and final model.

(iii) Regarding the necessity of the wavelet features. While deep learning models are considered to have the capacity to autonomously extract informative features from signals and may obtain some information provided by the wavelet features, several studies ^[2,3] have indicated deep learning model’s preference for low-frequency features. In our research, wavelet features can serve to address this aspect by offering information extracted from high-frequency features. This could potentially explain the advantageous impact of employing wavelets features. As shown in Figure. S1, the model with three input branches using “wavelet features” + “ECG signals” + “clinical features” generally outperforms the model with two input branches using “ECG signals” + “clinical features”, especially on recall and comprehensive metric ROC-AUC.

(iv) Regarding the overfitting accuracy, our model demonstrated notably performances on the internal test set and the two external test sets, which suggests strong generalization abilities and such results cannot solely be attributed to

overfitting.

References:

- [1] Hannun, Awni Y., Pranav Rajpurkar, Masoumeh Haghpanahi, Geoffrey H. Tison, Codie Bourn, Mintu P. Turakhia, and Andrew Y. Ng. "Cardiologist-level arrhythmia detection and classification in ambulatory electrocardiograms using a deep neural network." *Nature medicine* 25, no. 1 (2019): 65-69.
- [2] Tancik, Matthew, Pratul Srinivasan, Ben Mildenhall, Sara Fridovich-Keil, Nithin Raghavan, Utkarsh Singhal, Ravi Ramamoorthi, Jonathan Barron, and Ren Ng. "Fourier features let networks learn high frequency functions in low dimensional domains." *Advances in Neural Information Processing Systems* 33 (2020): 7537-7547.
- [3] Chen, Yunpeng, Haoqi Fan, Bing Xu, Zhicheng Yan, Yannis Kalantidis, Marcus Rohrbach, Shuicheng Yan, and Jiashi Feng. "Drop an octave: Reducing spatial redundancy in convolutional neural networks with octave convolution." In *Proceedings of the IEEE/CVF International Conference on Computer Vision* (2019): 3435-3444.

Thank you for your helpful comments again!

Reviewers' Comments:

Reviewer #1:

Remarks to the Author:

My concerns have been addressed. I thank the authors for their changes and congratulate them on their work.

Reviewer #3:

Remarks to the Author:

The authors have responded adequately to the questions and comments I raised. I have no additional comments. Thank you for the opportunity to review this valuable research.